# Spatially interacting phosphorylation sites and mutations in cancer

Kuan-lin Huang [1✉], Adam D. Scott[2], Daniel Cui Zhou [2], Liang-Bo Wang [2], Amila Weerasinghe [2], Abdulkadir Elmas [1], Ruiyang Liu[2], Yige Wu[2], Michael C. Wendl[2], Matthew A. Wyczalkowski [2], Jessika Baral[2], Sohini Sengupta[2], Chin-Wen Lai[3], Kelly Ruggles[4], Samuel H. Payne [5], Benjamin Raphael[6], David Fenyö [4], Ken Chen [7], Gordon Mills [8] & Li Ding[2✉]

Advances in mass-spectrometry have generated increasingly large-scale proteomics datasets containing tens of thousands of phosphorylation sites (phosphosites) that require prioritization. We develop a bioinformatics tool called HotPho and systematically discover 3D co-clustering of phosphosites and cancer mutations on protein structures. HotPho identifies 474 such hybrid clusters containing 1255 co-clustering phosphosites, including RET p.S904/Y928, the conserved HRAS/KRAS p.Y96, and IDH1 p.Y139/IDH2 p.Y179 that are adjacent to recurrent mutations on protein structures not found by linear proximity approaches. Hybrid clusters, enriched in histone and kinase domains, frequently include expression-associated mutations experimentally shown as activating and conferring genetic dependency. Approximately 300 co-clustering phosphosites are verified in patient samples of 5 cancer types or previously implicated in cancer, including CTNNB1 p.S29/Y30, EGFR p.S720, MAPK1 p.S142, and PTPN12 p.S275. In summary, systematic 3D clustering analysis highlights nearly 3,000 likely functional mutations and over 1000 cancer phosphosites for downstream investigation and evaluation of potential clinical relevance.

[1] Department of Genetics and Genomics, Tisch Cancer Institute, Icahn Institute for Data Science and Genomic Technology, Icahn School of Medicine at Mount Sinai, New York, NY, USA. [2] Department of Medicine, McDonnell Genome Institute, Department of Genetics, Siteman Cancer Center, Washington University in St. Louis, St. Louis, MO, USA. [3] Department of Pathology and Immunology, Washington University in St. Louis, St. Louis, MO, USA. [4] Center for Health Informatics and Bioinformatics, New York University School of Medicine, New York, NY, USA. [5] Department of Biology, Brigham Young University, Provo, UT, USA. [6] Lewis-Sigler Institute, Princeton University, Princeton, NJ, USA. [7] Departments of Bioinformatics and Computational Biology, The University of Texas MD Anderson Cancer Center, Houston, TX, USA. [8] Knight Cancer Institute, Oregon Health & Science University, Portland, OR, USA. ✉email: kuan-lin.huang@mssm.edu; lding@genome.wustl.edu

Dysregulated phosphorylation of oncogenic proteins alters pathway activity and contributes to tumor phenotypes[1,2]. Recent advances in mass-spectrometry have generated increasingly large-scale proteomics datasets in multiple cancer types[3,4], each containing tens of thousands of phosphosites that urgently require prioritization. Missense somatic mutations and phosphorylations, independently or through mutual interactions, can affect the physicochemical properties of the residue side chains and modulate protein functions or stability in oncogenic pathways. Thus far, mutation and phosphorylation have been largely studied in isolation by genomics and proteomics approaches. Integrated methodologies are required to reveal their interactions and prioritize both types of events with functional significance.

Previous works highlighted the potential functionality of mutations that are linearly adjacent to phosphosites in cancer driver genes[5–7], yet these studies did not consider the 3-dimensional structures of proteins. We and others previously demonstrated that mutations in cancer genes form 3-dimensional (3D) spatial clusters—defined by high local concentrations of mutations on protein structures—enriched for functional missense mutations[8–10]. We hypothesize that co-clustering mutations and phosphosites in spatial hotspots will also enrich for functional events of both categories. Systematic analyses of mutations from sequencing data and phosphosites from global proteomics data will enable us to investigate beyond currently-interrogated phosphosites with available targeting antibodies and reveal functionalities of phosphosites.

Here, we report on the development and application of a bioinformatics tool called HotPho to systematically discover spatial interactions of mutations and phosphosites. We find 474 significant hybrid clusters (defined as clusters containing both co-clustering phosphosites and mutations) that prioritize 1255 phosphosites and 2938 mutations on protein structures from large-scale proteomics and genomics data. Many co-clustering mutations are associated with high functional scores, expression changes, and known recurrent/activating events that expose genetic dependency; whereas many co-clustering phosphosites are found in kinase domains and verified in primary tumor samples. We specifically prioritize phosphosites co-clustering with activating mutations of BRAF, EGFR, and PIK3CA. Collectively, our approach of 3D spatial clustering on protein structures systematically highlights likely functional mutations and phosphosites for downstream investigation.

## Results

**HotPho algorithm and performance.** Extending beyond the originating framework of an earlier mutation-clustering tool we developed, namely HotSpot3D[8], HotPho enables investigation of proximal and structural information of phosphosites with their neighboring mutations and domains, both on a single protein structure or co-crystallized binding partners in a protein complex (Fig. 1). Briefly, all missense variants and phosphosites are considered as nodes and their 3D distances as edges on an undirected graph and the clusters are built up using the Floyd–Warshall shortest-paths algorithm implemented by HotSpot3D[8] ("Methods").

We demonstrated the capability of HotPho for identifying co-clustering cancer mutations and phosphosites using data comprising 225,151 unique phosphosites from PTMcosmos compiled from multiple databases and CPTAC cancer proteomic cohorts[3,4] ("Methods"). We also included 791,489 missense mutations from 9062 samples across 33 cancer types from a filtered set of Multi-Center Mutation Calling in Multiple Cancers project (MC3) mutation calls from the TCGA PanCanAtlas[11], taken in account their recurrence in the MC3 cohort. Both mutations and

phosphosites are mapped by HotPho and analyzed based on 5950 processed human proteins from UniProt[12] having at least one PDB structure.

To assess whether the co-clustering between aforementioned sets of mutations and phosphosites is non-random, we analyzed the clusters against a set of permutated data as follows: the original mutation backbone was maintained while phosphosites were randomly populated 100 times, keeping the corresponding ratios of residue types of phosphosites constant ("Methods"). We found a higher fraction of hybrid clusters in the original HotPho output (8.1%) at the top 5% of the cluster closeness score compared to the null distribution from the permutations (Fig. 1). We defined the criteria of high-confidence clusters to have cluster closeness scores within the top 5% of their respective cluster types and subsequently limited our analyses to these clusters. In hybrid clusters, the 5% sensitivity corresponded to 97.4% specificity in a receiver operating characteristic (ROC) curve analysis (AUC = 0.58, Supplementary Fig. 1A).

We conducted a multitude of analyses to investigate the modality in the score distribution and the implication of using the 5% threshold. First, while this threshold (cluster closeness score = 2.56) may permit false-positives if the simulated phosphosites only contain negatives, we observed many of the clusters containing activating or recurrent mutations with cluster closeness scores close to the threshold (Supplementary Data 1). It is possible that the spatial distribution of cancer mutations and commonly phosphorylated amino acid residues (i.e., serine, threonine, and tyrosine) is not random and thus retaining additional hybrid clusters is needed to minimize false-negatives. Second, to resolve possible reasons underlying the multi-modal distribution of cluster closeness scores, we compared the score distributions for 299 mutation-enriched cancer driver genes[13] versus other genes. While hybrid clusters involving driver genes showed a higher density at the higher-score mode, driver gene status did not guarantee high scores (Supplementary Fig. 1B). The 5% score threshold showed a sensitivity = 0.17 and specificity = 96.0% in distinguishing hybrid clusters with driver genes (Supplementary Fig. 1C). Finally, we examined the score distribution using 200 bins on both the simulated vs. observed clusters, finding multiple peaks and alternative thresholds, for example, thresholding using one of the higher local minima retained only the top 2.28%, or the top 216 clusters (Supplementary Fig. 1D). Cluster closeness scores for all identified clusters are provided herein to prioritize a more stringent set of clusters (Supplementary Data 1).

**Co-clustering of phosphosites and mutations using HotPho.** HotPho generated a final high-confidence set of 906 mutation-only, 127 phosphosite-only, and 474 hybrid clusters based on the top 5% cluster closeness score threshold ("Methods", Supplementary Data 1). Top genes harboring each type of cluster varied (Fig. 2a): MGAM, SI, ERBB3, and LRRC4C each had at least 9 mutation-only clusters and such type of clusters have been previously characterized[8–10]. Phosphosite-only clusters are found in fewer instances: ANXA5, CLIP1, FLNB, GPI, HSPD1, PEBP1, and PTK2 each harbored two (Supplementary Fig. 2A).

For subsequent analyses, we focused on investigating hybrid clusters found across 474 unique proteins (some proteins only form hybrid clusters with their protein complex partners). Notably, the highest counts of hybrid clusters were found for genes known for recurrent mutations, including TP53 (10 hybrid clusters), PIK3CA (8), CTNNB1 (6), EGFR (6), and other genes involved in cancers, such as HIST1H2BC (6) and PLG (5) (Fig. 2a). These clusters comprise a total of 1255 phosphosites and 2938 mutations. The composition of

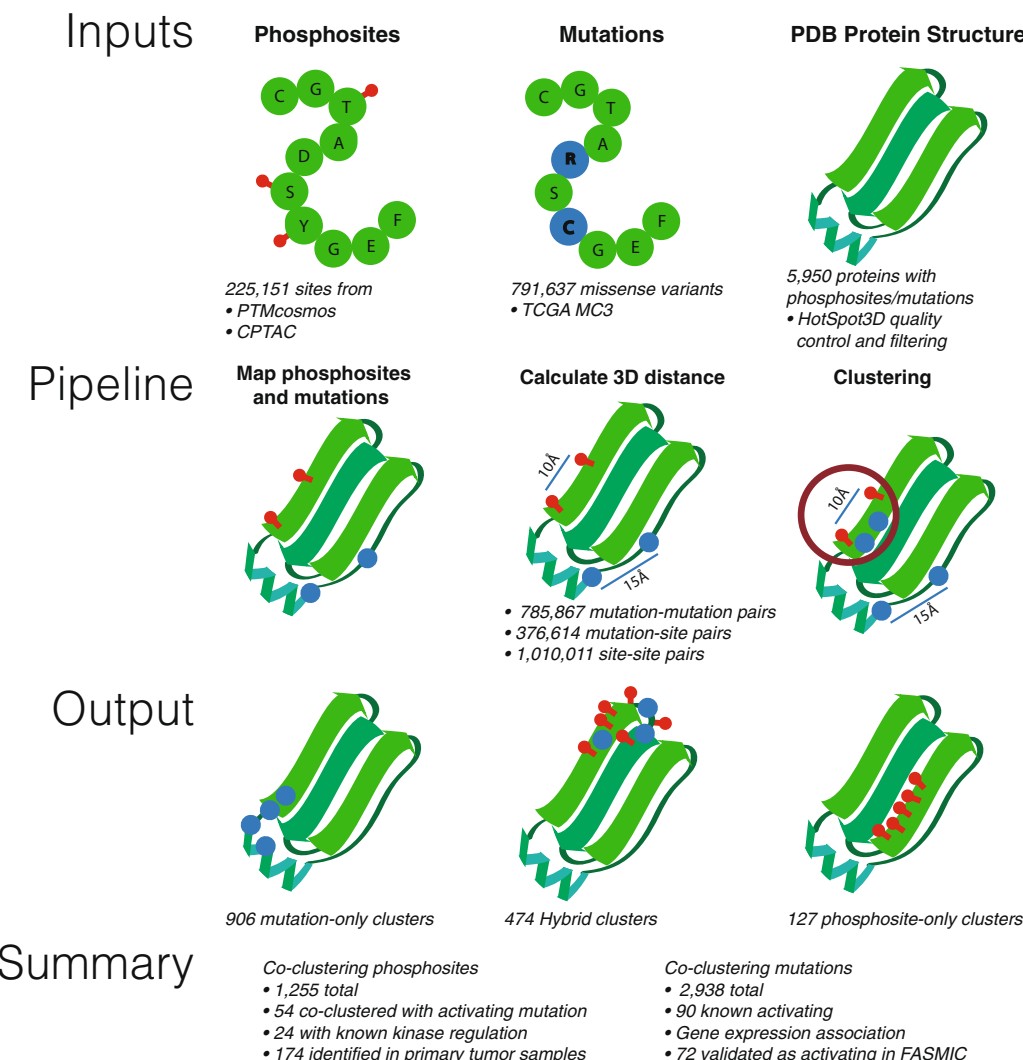

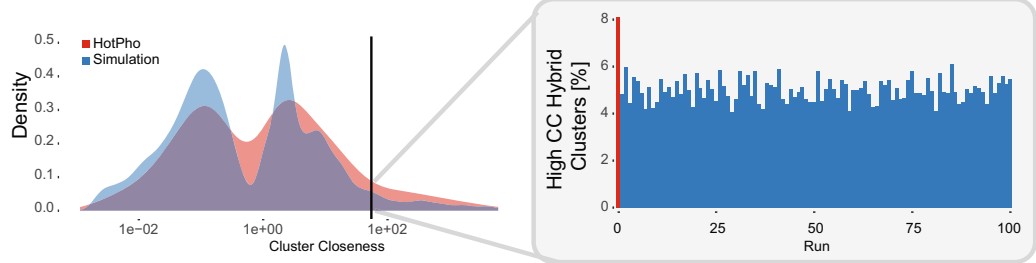

**Fig. 1 HotPho workflow and performance benchmarks. a** HotPho takes user-provided lists of mutations and phosphosites as inputs, map them onto PDB protein structures, calculates each of the pairwise distances, conducts clustering, and reports clusters with prioritized mutations and phosphosites. **b** Comparison of HotPho results measured using phosphosite and MC3 cancer mutation data vs. simulated data of randomly distributed phosphosites. The left panel indicates the density of cluster closeness (CC) scores for all hybrid clusters in the HotPho run and the simulated runs, where the vertical line indicates the top 5% score threshold. The right panel shows the bar plot comparing the number of hybrid clusters passing the same 5% threshold in the HotPho and simulated runs.

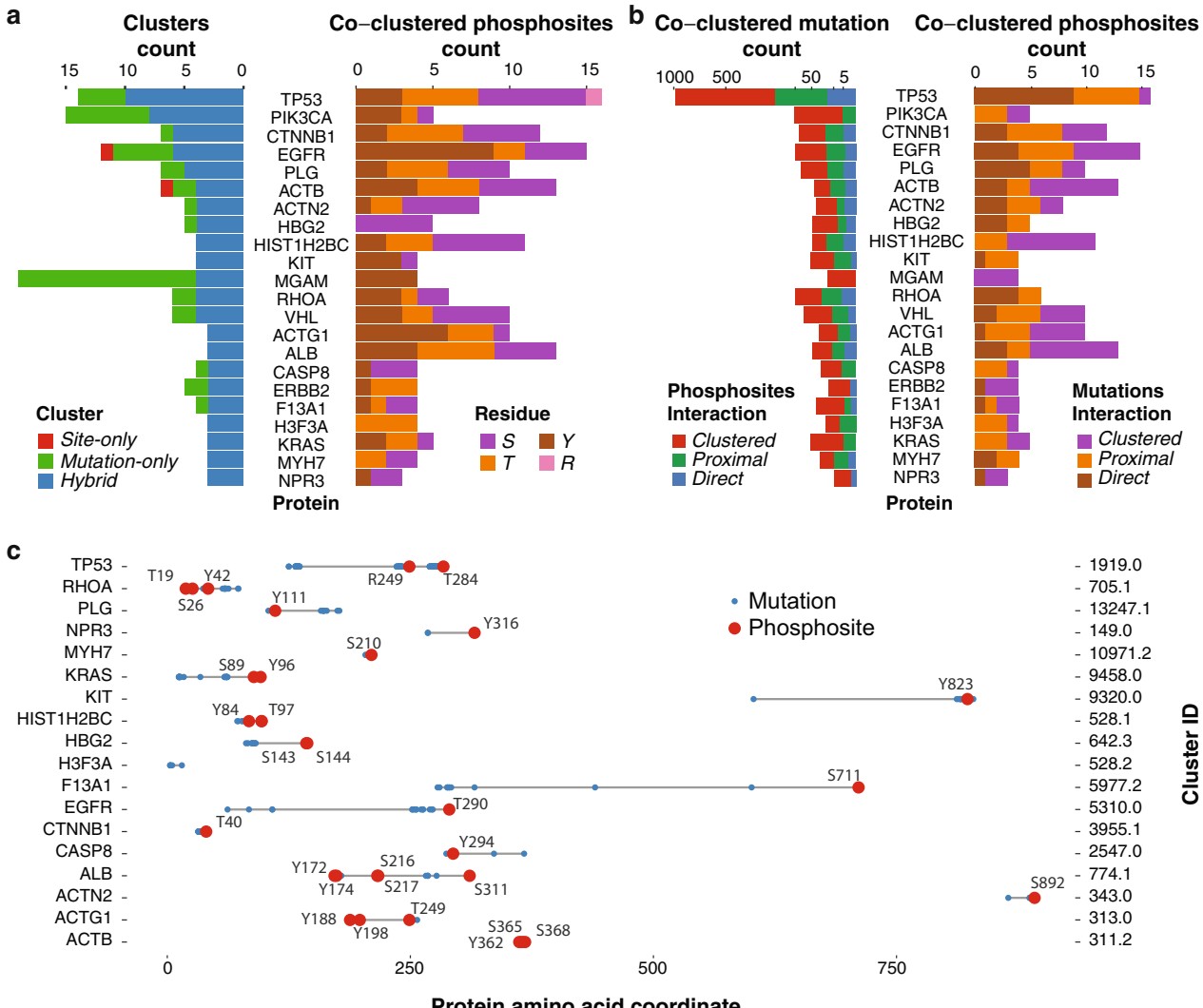

**Fig. 2 Hybrid clusters containing both phosphosites and mutations. a** Left shows the counts of hybrid clusters, mutation-only clusters, and site-only clusters in genes with at least two hybrid clusters. Right barplot shows counts of each type of phospho-residue, being serine (S), threonine (T), tyrosine (Y), or Arginine (R), found in hybrid clusters for each of the genes. **b** Spatial interactions of co-clustered mutations and phosphosites. For each of the genes, we counted how many of the co-clustered mutations and phosphosites are also directly overlapping (Direct) or within 2 amino acid residues (Proximal), and without any of these apparent linear relationships (Clustered) to phosphosites and mutations, respectively, in the same hybrid clusters. **c** Phosphosite and mutations on the linear protein coordinate of top hybrid clusters as defined by cluster closeness scores in each of the highlighted genes (excluding HIST1H4G and H3F3A due to their top hybrid clusters included phosphosites from other proteins and excluding MGAM due to its top clusters located at residue coordinate beyond the plotted range [centroid at 1514]). The Ensembl transcripts used for mapping of the protein coordinates are described in Supplementary Data 1.

co-clustered phosphosites differs across gene products; co-clustered tyrosines are most commonly observed in PIK3CA and EGFR kinases, whereas serines are most common in HIST1H2BC and HBG2 (Fig. 2a). The top hybrid clusters of each protein—identified by the highest cluster closeness score—may span mutations and phosphosites that are far from one another in the linear distance (Fig. 2c, Supplementary Fig. 2B). Phosphosites prioritized in these clusters include CTNNB1 p.T40, EGFR p.T290, ERBB2 p.T733/T759, KIT p.Y578, and TP53 p.T284. We also compared the mutations in the hybrid clusters to those found in a clustering analysis using only TCGA MC3 mutations, which contained 9403 clustered mutations. Among the 2938 mutations found in the 474 hybrid clusters, we found only 48 mutations not found by mutation-only clustering. The list of 48 mutations contained mutations of interest in PDE1B (5 mutations), SRSF7 (4 mutations), and PTPN12 p.S275F/C that

co-localized with p.S275 and co-clustered with p.S39/p.T40 (Supplementary Data 2).

Among the 1,255 co-clustered phosphosites, 291 sites directly overlap and 356 sites are proximal (within 2 amino acid residues linearly) to their co-clustered mutations (Supplementary Data 1, Fig. 2b). The HotPho co-clustering analysis adds a substantial count of 608 phosphosites which are distant in terms of a linear sequence, yet close in 3D protein structure, including the majority of the sites found on ACTB, HIST1H2BC, and ERBB2. Nearly half of the clusters we identified can only be found by integrating 3D protein structure, demonstrating the added value of 3D approaches for the discovery of spatial relationships between mutations and phosphosites.

We then examined whether proteins containing hybrid clusters are enriched in specific biological pathways curated by WikiPathways[14] and the NCI-Nature Pathway Interaction

Database[15] using Enrichr[16] (Supplementary Data 3, Supplementary Fig. 3, "Methods"). The most enriched NCI-Nature pathways include PDGFR-beta, ErbB2/ErbB3, ErbB1, hepatocyte growth factor receptor (c-Met), SHP2, Fc-epsilon receptor I, and mTOR signaling pathways (Fisher's exact test, adjusted $P < 1E-12$), which is reaffirmed by the Focal Adhesion-PI3K–Akt–mTOR-signaling pathway being one of the top enriched WikiPathways (adjusted $P = 8.8E-16$). These findings suggest the possible involvement of hybrid clusters and co-clustering phosphosites in oncogenic signaling pathways.

We further hypothesized that hybrid clusters would be enriched in functional domains related to oncogenic processes[17]. Mapping residues to PFAM domains, we identified 26 PFAM domains significantly enriched for mutations and phosphosites in hybrid clusters when comparing to the background of all mapped mutations and phosphosites (Supplementary Data 4, Supplementary Fig. 3, "Methods"). Domains of histone proteins, including centromere protein Scm3, core histone H2A/H2B/H3/H4, and centromere kinetochore component CENP-T histone fold showed the most significant enrichment (Fisher's exact test, FDR ≤ 1.53E−43). Another two top PFAM domains are protein tyrosine kinase and protein kinase domains (Supplementary Fig. 3). Specifically, we identified hybrid clusters in tyrosine kinase domains of tyrosine kinases (TK), such as MET, FGFR2/3, and ERBB2/3, and in BRAF of the tyrosine kinase-like (TKL) group. Other hybrid clusters involving sites at protein kinase domains included TGFBR1 of the TKL group, MAP2K4 of the STE group, and AKT1/2 of the AGC group. Notably, some kinase-domain clusters showed conserved mutation/phosphosite patterns across homologs, such as FGFR2 and FGFR3 (Supplementary Fig. 3).

**Co-clustering phosphosites adjacent to known activating mutations**. To prioritize candidate phosphosites, we first investigated phosphosites co-clustering with known functional cancer mutations. We curated experimentally validated mutations from the Cancer Biomarkers database with Cancer Genome Interpreter[18], OncoKB[19], and KinDriver[20], collecting a total of 367 activating mutations ("Methods"). We found 29 hybrid clusters containing 90 of these activating mutations in 17 genes, suggesting the functional relevance of the 54 co-clustering phosphosites (Supplementary Data 5). PIK3CA and EGFR are each involved in 4 hybrid clusters containing activating mutations and such clusters are also found in CTNNB1 (3), KIT (3), BRAF (2), ERBB2 (2), KRAS (2), MET (2), and NRAS (2).

Phosphosites co-clustering with activating mutations are likely of functional relevance. We specifically examined these clusters on protein structures (Fig. 3b, Supplementary Fig. 5). Both ERBB2 p.T733 and p.T759 are located adjacently to the activating mutation p.L755W. NRAS phosphosite p.Y64 is co-clustered with two of the most recurrently mutated residues p.G12 and p.Q61. Receptor tyrosine kinases, KIT, MET, and RET all harbor phosphorylated tyrosine sites co-clustering with activating mutations. These prioritized phosphosites include KIT p.Y578, MET p.Y1093/Y1159/Y1230, and RET p.Y928. Two hybrid clusters containing activating mutations were found on a protein complex formed by PIK3CA/PIK3R1: PIK3R1 phospho-tyrosines p.Y470 and p.Y556 clustered with activating mutations PIK3CA p.N344G/M, p.N345K, p.C420R, and PIK3R1 p.N564D. In the other hybrid cluster, PIK3R1 p.T463 clustered with activating mutations PIK3CA p.E453K/Q. The co-clustering phosphosites next to known activating mutations are promising targets for further investigation, along with their adjacent mutations.

We hypothesized that phosphosites co-clustering with highly-recurrent mutations in a cancer cohort might imply functionality in the specific cancer type. We calculated the frequency of each of the co-clustering mutations within each of the TCGA cancer cohorts and identified their spatially adjacent phosphosites (Supplementary Data 6). We found that co-clustering phosphosites of the most recurrent mutations aggregate in proteins, including CTNNB1, HRAS, IDH1, KRAS, NRAS, PIK3CA, and TP53 (Fig. 3a, Supplementary Fig. 5). In PIK3CA, we identified p.T957 co-clustering with the highly recurrent p. H1047R that affects many gynecologic cancer cases, including 13.8% of breast invasive carcinoma (BRCA), 7% of uterine carcinosarcoma (UCS), and 5.8% of uterine corpus endometrial carcinoma (UCEC). In TP53, phosphosites p.R249 and p.T284 co-cluster with p.R273C/H that affects 11.4% of brain lower-grade glioma (LGG), 5.3% of UCS, and 3.8% of esophageal carcinoma (ESCA); TP53 p.T155 and P.S183 co-cluster with p. R175H that affects 8.3% of rectum adenocarcinoma (READ), 6.3% of colon adenocarcinoma (COAD), 6% of ESCA, 3.7% of ovarian serous cystadenocarcinoma (OV), and 3.5% of UCS.

Many phosphosites co-clustering with recurrent mutations were found in protein homologs. IDH1 phosphosites p.Y135 and p.Y139 co-clustered with p.R132H, which is highly recurrent in brain tumors (73.6% of LGG and 6.1% of glioblastoma multiforme [GBM]), as well as p.R132C implicated in several cancer types (17.1% of cholangiocarcinoma [CHOL], 4.3% of acute myeloid leukemia [LAML], 3.4% of LGG, and 3.2% of skin cutaneous melanoma [SKCM]). In its homolog protein IDH2, p.Y179 co-clustered with p.R140Q affecting 6.5% of LAML (Fig. 3). For the Ras proteins, KRAS/HRAS/NRAS all harbor highly recurrent mutations for residues p.G12/G13 that affect large fractions of pancreatic adenocarcinoma (PAAD), COAD, READ, lung adenocarcinoma (LUAD), and UCEC. Each harbors overlapping yet distinct sets of co-clustering phosphosites—KRAS p.S89/p.Y96, NRAS p.Y64, and HRAS p.Y32/T35/Y64/Y96—warranting further investigation into their potentially shared and distinct signaling functions across cancer types (Fig. 3).

**Functional evidence for co-clustering mutations**. We evaluated whether HotPho can effectively prioritize functional mutations in hybrid clusters by comparing with functional scores predicted by VEST[21], Mutation Assessor[22], PolyPhen2[23], SIFT[24], and a composite Eigen score composed of all these scores[25] (Fig. 4a). Within proteins harboring hybrid clusters, clustered mutations showed strikingly higher predicted functional scores compared to other mutations in the same proteins (Wilcoxon rank-sum test, $P < 2.2e-16$), supporting the view that co-clustered mutations should be prioritized.

To further demonstrate that hybrid clusters enrich for functional mutations, we examined whether clustered mutations are associated with the protein or phosphoprotein changes, as previously found for functional and pathogenic mutations[26,27]. Using the TCGA Reverse Phase Protein Arrays (RPPA) dataset for each of the 33 cancer types, we conducted a differential expression analysis to search for protein/phosphoprotein markers expressed at different levels in carriers of clustered mutations ("Methods"), identifying 24 significant (FDR < 0.05, linear regression) gene-cancer associations (Fig. 4b, Supplementary Data 7). *TP53* mutations in hybrid clusters are significantly correlated with higher p53 protein expression in 14 cancer types, most strikingly in UCEC, BRCA, COAD, and OV, consistent with the previously reported *cis*-effect of functional TP53 missense mutations[28]. Clustered EGFR mutations are likewise associated with higher EGFR protein and EGFR p.Y1068 expression in LGG, GBM, and LUAD, cancer types largely affected by activating mutations of *EGFR*. We also found clustered *KIT* mutations to be associated with higher c-Kit in TGCT and SKCM (Fig. 5b).

At a single residue level, we noted clustered mutations showing high protein or phosphoprotein expressions above the 95th

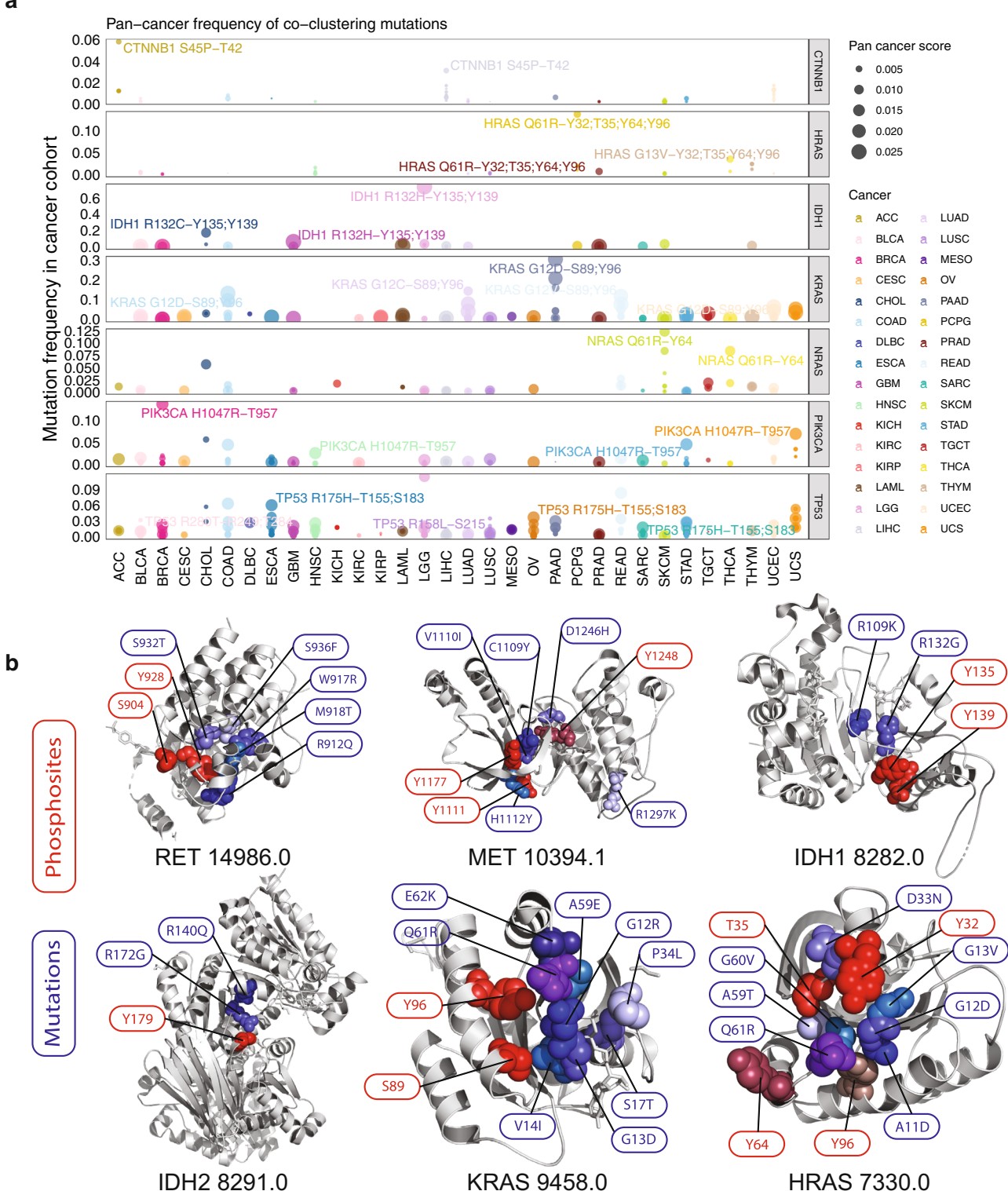

**Fig. 3 Hybrid clusters highlighting co-clustering phosphosites adjacent to recurrent mutations across cancer types. a** The frequency of co-clustering mutations within each TCGA cancer type. For each of the cancer types shown in a distinct color, the mutation with the highest recurrence is labeled along with its co-clustering phosphosites. The size of the dot to indicate the pan-cancer score calculated by averaging the frequencies of the mutation across the 32 cancer types in TCGA. **b** Selected hybrid clusters with activating mutations shown on 3D protein structures obtained through PDB. Mutations are colored in shades of blue and phosphosites are colored in shades of red.

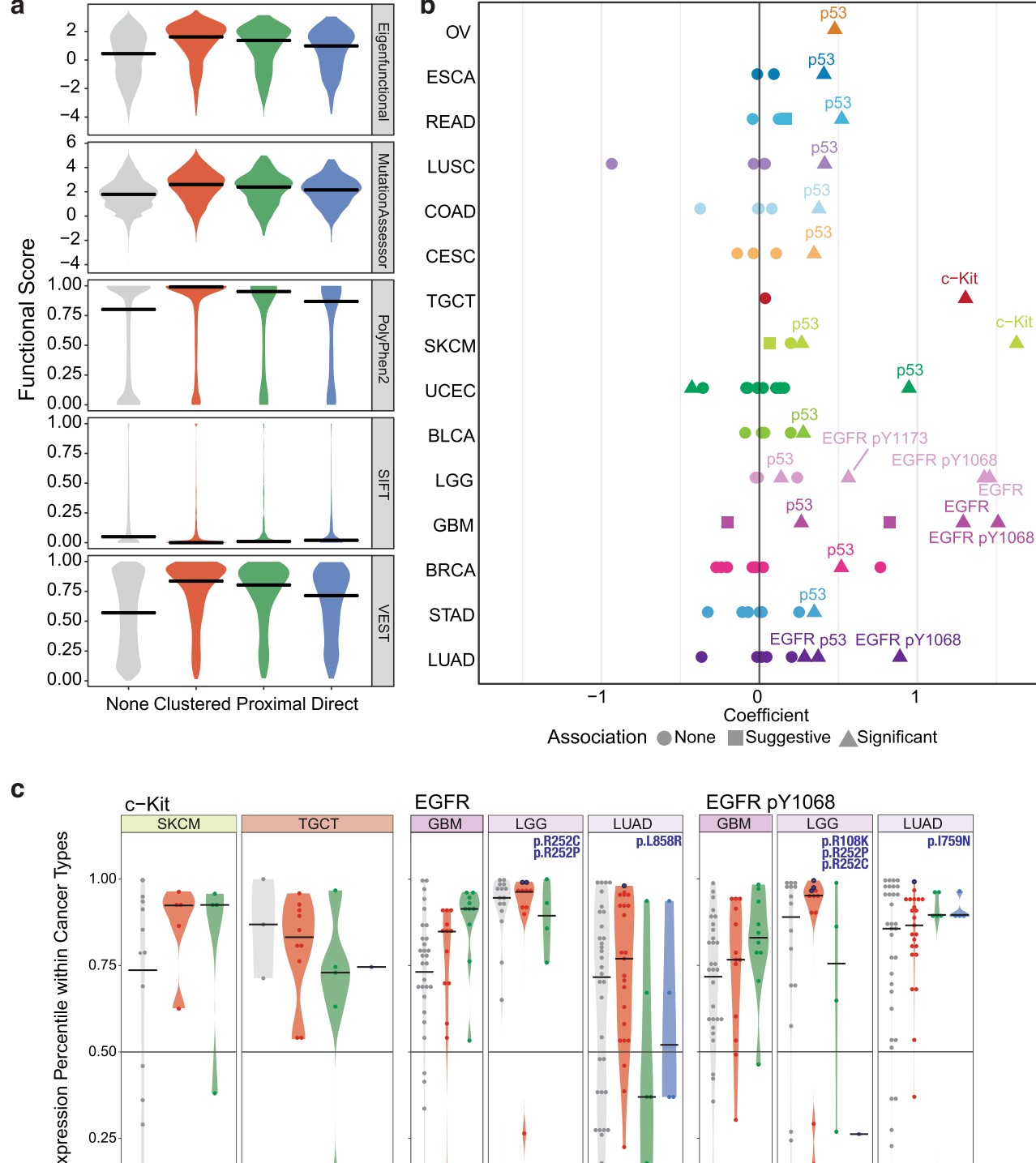

**Fig. 4 Proteomic effects associated with co-clustered mutations. a** Comparison of predicted functional scores, including those provided by Mutation Assessor, PolyPhen-2, SIFT, VEST, and an eigen score for mutations having different spatial interactions with phosphosites. The interaction types are direct (directly overlapping), proximal (within 2 residues in the linear distance), clustered (in hybrid clusters), and none of these interactions. **b** Plot showing cancer types where the carrier of each gene's co-clustered mutation is associated with significantly higher or lower expression of the corresponding protein/phosphoprotein marker in the Reverse Phase Protein Arrays (RPPA) dataset. Each dot represents a gene-cancer association, where color depicts cancer type and shape shows significance. **c** Expression percentile of the protein/phosphoprotein marker in carriers of multiple types of mutations (direct, proximal, clustered, none) in the corresponding genes. Mutations carried by the samples with greater than 97% marker expression are further text-labeled.

percentile in the same cancer cohort (Fig. 4c). These include recurrent TP53 mutations p.R248Q/W, p.R273H/C/L, p.R175H, p.R282W/G, and p.R337C. Top mutations in EGFR differ between brain and lung tumors: in LGG, EGFR p.R108K, p.R252C/P, and p.R263I, which are adjacent to the phospho-threonine p.T290, are associated with high EGFR protein and phosphoproteins. Many samples with top EGFR expression in GBM also carry mutations in the same hybrid cluster, p.A289V/T and p.R252C/P, whereas in LUAD the associated mutation is the recurrent EGFR p.L858R (4.3% of LUAD) co-clustering with phosphosites p.Y869 and p.Y891. KIT mutation p.D816V and p.829P is associated with high c-kit and it clustered with p.Y362 and p.Y823 (Fig. 4c).

To further validate these findings, we conducted similar analyses of the mutational impact on protein expression using global proteomics datasets from retrospective and prospective CPTAC cohorts of breast, ovarian, and colorectal cancers (2 cohorts/cancer) each comprised of 78–126 samples ("Methods", Supplementary Data 8, Supplementary Fig. 6). Given the limited sample sizes, no associations passed multiple testing thresholds (FDR < 0.05) and only suggestive associations (Wilcoxon rank-sum test, $P < 0.05$) are highlighted herein: we validated that TP53 co-clustering mutations associated with higher protein expression in all three cancer types. In colorectal cancer, KRAS mutations in cluster 9458.0 affecting p.G12, p.G13, p.V14, and p.Q61 are associated with higher KRAS expression, whereas HNF4A mutations in cluster 7977.0 are associated with low HNF4A expression. Other notable findings include that ESR1 mutations in cluster 1357.1 are associated with high ESR1 in ovarian cancer, whereas AKT1 mutations in cluster 756.0 (Supplementary Fig. 6) are associated with high AKT1 proteins.

Finally, given the potential functionality of co-clustering mutations, we characterized the mutational landscape across cancer types in the TCGA MC3 dataset of ~10 thousand tumors[11]. By considering missense mutations (1) directly overlapping phosphosites, (2) proximal to phosphosites, and (3) co-clustering with phosphosites, we noticed that considering co-clustering mutations contribute significantly to the fraction of potentially functional mutations in many cancer genes including EGFR, KRAS, and PIK3CA (Supplementary Fig. 7).

**Functional verification of co-clustering mutations**. We next tested whether hybrid clusters were enriched for functional mutations, including those shown to be activating and confer genetic dependency by cancer cells. To test for activating mutations that confer clonal selection advantages, we assessed experimental data from 1054 somatic mutations in the Ba/F3 and MCF10A cells[29], including 549 found in genes with hybrid clusters. Out of the 549 unique somatic mutations, 86 co-clustered with phosphosites and 463 did not. There was a striking enrichment of activation in hybrid mutations co-clustering with phosphosites. For mutations functionally assessed in Ba/F3, 77.6% (66/85) of the co-clustering mutations were determined as activating compared to only 30.2% (138/457) of the other mutations determined as activating (One-tailed Fisher's exact test, $P = 2.66E{-}16$, Fig. 5a). For mutations functionally assessed in MCF10A, 67.6% (46/68) of the co-clustering mutations were activating compared to only 35.2% (146/415) of the other mutations being activating ($P = 5.03E{-}7$, Fig. 5b). Collectively 72 co-clustered phosphosites were determined as activating in either Ba/F3 or MCF10A.

We then examined whether the co-clustered mutations show significant enrichment of activating mutations compared to the other mutations on a gene-by-gene basis (Fig. 5a, b, Supplementary Data 9). Co-clustering mutations of PIK3CA are significantly

enriched for activating events in both cell lines (One-tailed Fisher's exact test, $P \leq 1.22E{-}3$), with 33/35 co-clustering mutations being validated in Ba/F3 and 21/22 in MCF10A; its binding partner PIK3R1 also shows suggestive enrichment in Ba/F3 ($P = 0.072$, 3/3). Significant enrichment of activating mutations was also observed for EGFR ($P = 1.63E{-}4$, 10/11) and BRAF ($P = 1.11E{-}3$, 12/18) co-clustering mutations in Ba/F3. In MCF10A, we also noted suggestive associations for BRAF, where 7/18 co-clustering mutations are activating ($P = 0.068$), and ESR1, where 7/18 co-clustering mutations are activating ($P = 0.068$). The enrichment of activating mutations in hybrid clusters suggests structural adjacency to phosphosites implies functional significance in oncogenes.

To evaluate the added predictive power of mutation functionality provided by phosphosite co-clustering, we examined the relationships between mutation functionality versus co-clustering mutation counts, mutation recurrence, and co-clustering with phosphosites (Fig. 5c, d). First, using a multivariate logistic regression model corrected for the mutated gene, we found that the number of co-clustering mutations was not significantly associated with the mutation functionality in either the BAF3 ($P = 0.91$) or MCF ($P = 0.40$) cell line (Supplementary Fig. 9A). Second, using a multivariate logistic regression model corrected for the mutated gene, we found that the recurrence of mutations in the TCGA MC3 cohort was significantly associated with the mutation functionality in both the BAF3 ($P = 2.8e{-}3$) or MCF10A ($P = 0.023$) cell lines. But, when adding the phosphosite co-clustering status to the regression model, the mutation functionality was no longer associated with recurrence ($P > 0.21$), but strongly associated with the co-clustering status in both BAF3 ($P = 1.4e{-}10$) and MCF10A ($P = 1.5e{-}4$) cell lines (Fig. 5c, d, Supplementary Fig. 9A). Altogether, these results suggest that spatial co-clustering with phosphosites may improve the prediction of mutation functionality beyond the commonly used mutation recurrence.

Next, we sought to test whether the co-clustered mutations may confer genetic dependency to the mutated cancer cells. In this case, cancer cells with co-clustered mutations would show higher vulnerability in a CRISPR-knockout screen targeting the mutated genes than cells with other mutations. To test this hypothesis, we utilized data using characterized by the CRISPR-knockout screens in the Cancer Dependency Map (DepMap) project[30], where a negative CERES dependency score indicates genetic dependency of the cancer cell. Within each of the 27 tested lineages, we carried out a Wilcoxon rank-sum test between the cell lines with co-clustered missense mutations versus those with other missense mutations ("Methods"). Strikingly, cancer cell lines with co-clustered mutations showed significantly higher dependency (or more vulnerability upon genetic knockout) than those with missenses in 14 lineages (Wilcoxon rank-sum test, FDR < 0.05), most notably lung, colorectal, skin, pancreas, and gastric cancer cells (FDR $\leq 3.3E{-}7$, Fig. 5e). We also obtained similar results when comparing cell lines with co-clustered missense mutations versus other non-synonymous mutations (Supplementary Data 10). Overall, these analyses showed that co-clustered mutations adjacent to phosphosites are enriched for activating events and highlight the genetic vulnerability of cancer cells.

**Verification of co-clustering phosphosites**. To verify co-clustering phosphosites, we sought evidence of these events being observed in the CPTAC proteomic cohorts of prospective primary tumor samples of 123 breast invasive carcinoma (BRCA), 83 ovarian carcinoma (OV), 97 colorectal adenocarcinoma (CRC), 103 uterine corpus endometrial carcinoma (UCEC), and

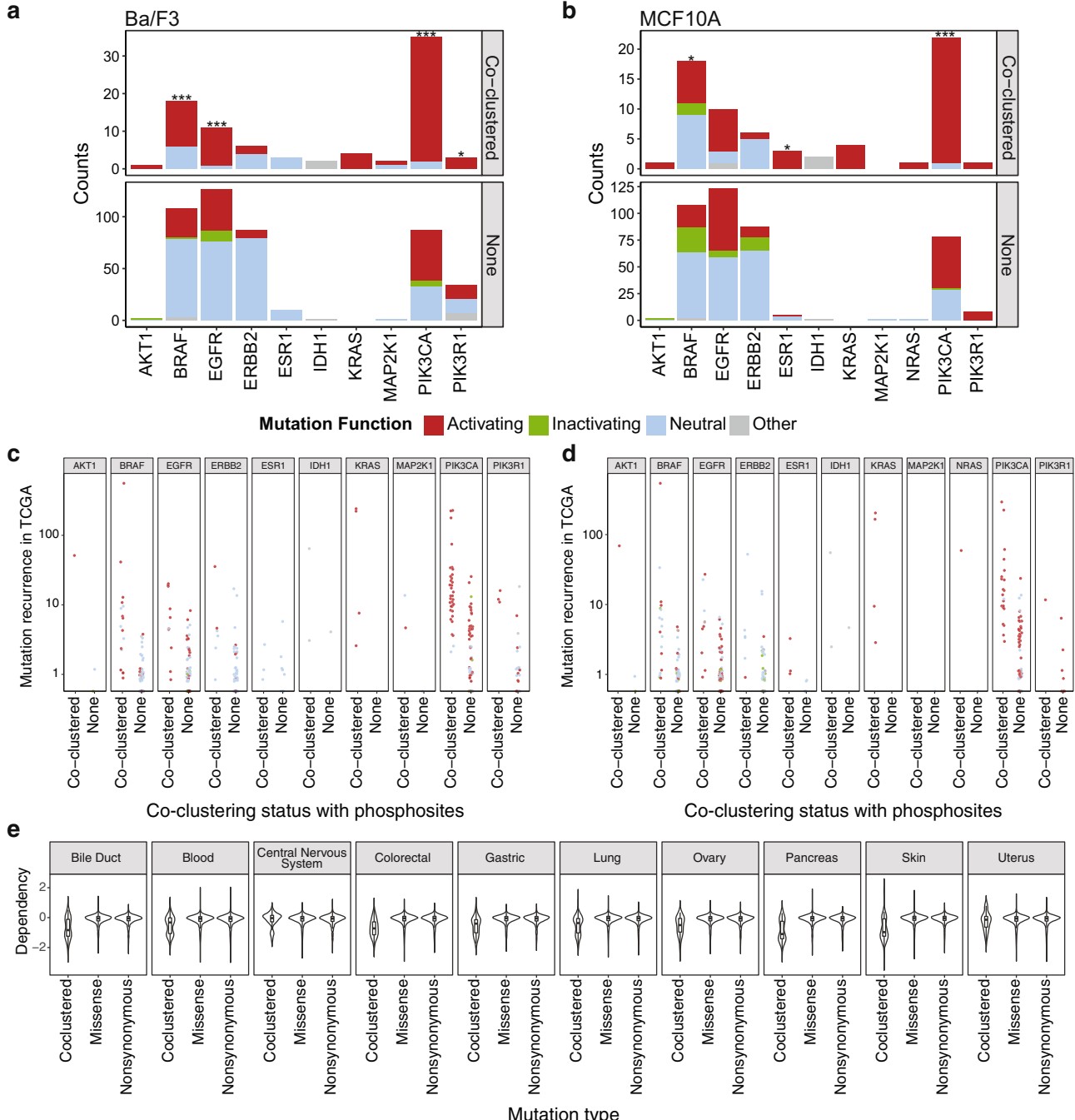

**Fig. 5 Functional assessment of co-clustering mutations.** Experimental validation data of co-clustering somatic mutations were extracted from previous systematic assessments in **a** Ba/F3 and **b** MCF10A cell lines[29]; we evaluated the functionality of 1054 somatic mutations in a cell viability assay, where each of the evaluated mutations were assessed through one transfected cell colony compared to control cell colonies. The number of activating mutations vs. other types of mutations in each gene was then calculated for the set of co-clustering mutations adjacent to phosphosites as discovered by HotPho and other non-clustering mutations. The asterisk indicates the significance of the association (One-tailed Fisher's exact test, ***$P < 0.005$, ** $0.005 <= p < 0.05$, * $0.05 <= p < 0.1$). The gene products showing significant associations include EGFR ($P = 1.63E-4$) and BRAF ($P = 1.11E-3$) co-clustering mutations in Ba/F3, and PIK3CA in Ba/F3 ($P = 1.67E-5$) and MCF10A ($P = 0.0012$). At a single mutation level, the functional status (color codes) between co-clustered versus other mutations are further shown for **c** Ba/F3 and **d** MCF10A cell lines against the mutation recurrence on the Y-axis, demonstrating the additional predictive power of co-clustering status on mutation functionality. **e** Dependency CERES score comparison of cell lines with co-clustered vs. other missense vs. other non-synonomous mutations in the DepMap CRISPR screen dataset. The 10 lineages with the highest numbers of co-clustered mutations are shown, including cell lines of the bile duct ($N = 30$), blood ($N = 44$), central nervous system ($N = 61$), colorectal ($N = 36$), gastric ($N = 27$), lung ($N = 107$), ovary ($N = 43$), pancreas ($N = 34$), skin ($N = 54$), and uterus ($N = 26$) tissues. The centre line corresponds to the median; lower and upper hinges correspond to the first and third quartiles (the 25th and 75th percentiles), respectively; the whiskers extend from the hinges to the largest value no further than 1.5 * IQR from the respective hinge, where IQR (inter-quartile range) is the distance between the first and third quartiles.

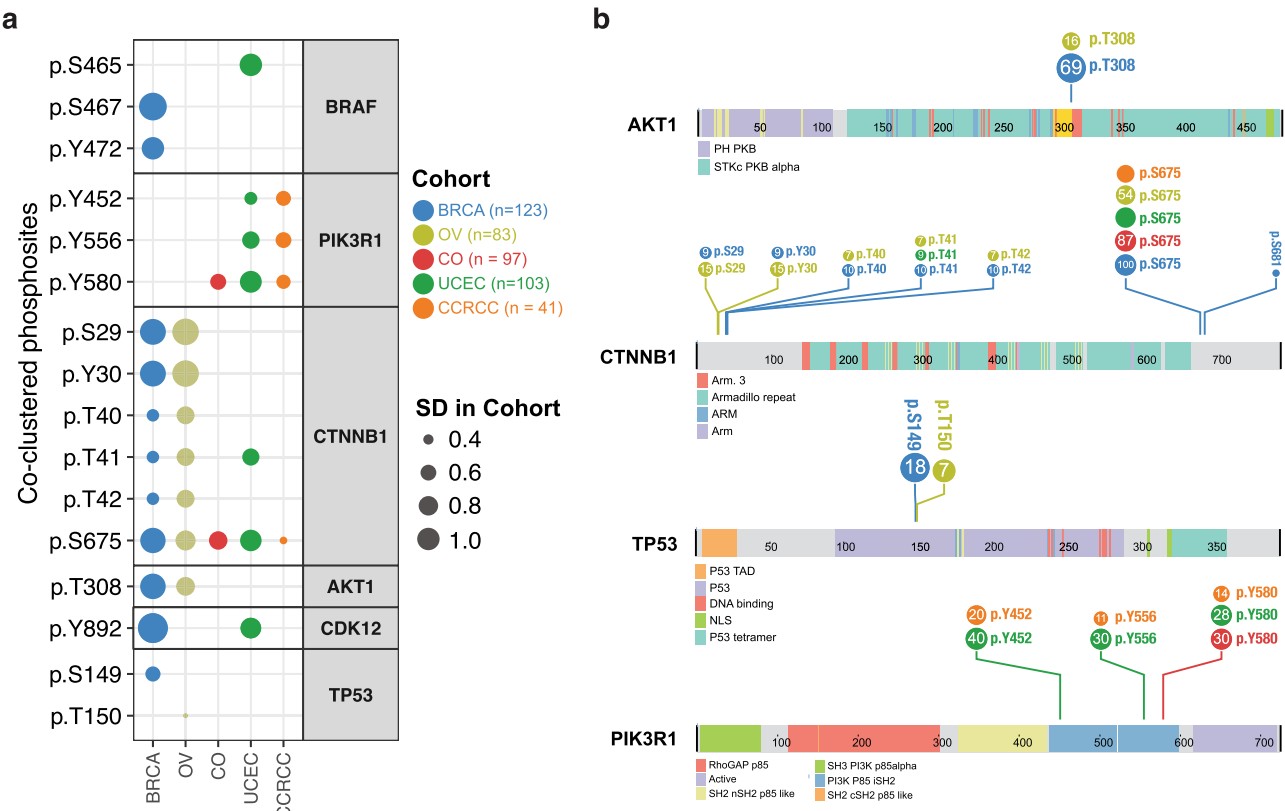

**Fig. 6 Verification of co-clustering phosphosites in primary tumors. a** Verification of co-clustering phosphosites of cancer proteins in patient tumor samples characterized by the CPTAC prospective projects. Each dot represents the phosphosites being detected in the cancer cohort colored-coded by cancer type. The size of the dot represents the standard deviation of the phosphosite level in the respective cancer cohort. **b** Lolliplots showing number of tumor samples (number in circle) where the co-clustering phosphosites of AKT1, CTNNB1, TP53, and PIK3R1 proteins are detected in each cancer cohort out of 123 breast invasive carcinoma (BRCA), 83 ovarian carcinoma (OV), 97 colorectal adenocarcinoma (CRC), 103 uterine corpus endometrial carcinoma (UCEC), and 41 clear cell renal cell carcinoma (CCRCC).

41 clear cell renal cell carcinoma (CCRCC). Of the 1255 co-clustered phosphosites, 259 were detected in at least one of the 5 cohorts (Fig. 6a, Supplementary Data 11). Some phosphosites may be cancer-type specific: we uniquely observed BRAF p.S467/Y472 in BRCA and p.S465 in UCEC. For TP53, we uniquely observed p.S149 in breast cancer versus TP53 p.T150 in ovarian cancer. Other phosphosites are found in multiple cancer types (Fig. 6b); for instance, AKT1 p.T308 was seen in both breast and ovarian cancers. Notably, phosphosites (p.S29/Y30/T40/T41/T42) near the section-terminus of the CTNNB1 protein were commonly seen in breast and ovarian cancers (p.T41 was also observed in UCEC), and p.S675 was detected in substantial numbers of samples in all 5 cancers (Fig. 6b). Co-clustering tyrosine phosphosites, PIK3R1 p.Y452/556/580, on the other hand, were observed in endometrial and renal cancer (p.Y580 was also observed in colorectal cancer). As a cautionary note, given the different reference samples and mass spectrometry runs for each cancer cohort, the cancer-specific phosphosites observed herein require further validation. Nonetheless, the detection of the co-clustering phosphosites in primary tumor samples further implicates their functionality in oncogenesis.

Finally, we conducted a systematic literature review of co-clustering phosphosites regulated or implicated in cancer ("Methods"), finding 25 unique phosphosites across 18 proteins that were experimentally linked to cancer (Supplementary Data 12). These include AKT1 p.T308 and CTNNB1 p.T41 found in CPTAC patient tumors, as well as sites with known kinase regulations, including EGFR p.S768/Y869/Y1016, ESR1 p.Y537, and TP53 p.S376/378. Other co-clustering phosphosites

showing functionality related to cancer include: MAPK1 p.S142 that was previously shown to be critical to the ERK2 docking domain and its mutated form p.S142L confers gain-of-function[31]; CTCF p.T374 that, along with a few nearby residues, were shown to be phosphorylated during mitosis and to decrease its DNA-binding activity[32]; BRAF p.T599 and p.S602 that are conserved from *C. elegans* to mammals and required for activation of the B-Raf kinase[33,34], and RB1 (Rb) p.S567 that is uniquely phosphorylated by MAPK11 (p38), triggering Rb-Hdm2 interaction and apoptosis[35]. These findings further validate the functionality of selected co-clustering phosphosites HotPho identified and suggest other sites in hybrid clusters may be prioritized for downstream investigations.

## Discussion

We describe the first systematic discovery of co-clustering mutations and phosphosites on 3D protein structures, a feat enabled by a bioinformatics tool—HotPho. HotPho successfully identifies likely functional mutational clusters and phosphosites in known cancer proteins, including EGFR, KIT, and KRAS/HRAS/NRAS, many of which are in kinase domains (Fig. 2). Co-clustering mutations in these clusters have higher predicted functional scores, increased protein/phosphoprotein levels (Fig. 4), and are experimentally validated as being functional and confer genetic dependency by cancer cells (Fig. 5). Concurrently, co-clustering phosphosites show multiple characteristics supporting their contributions in oncogenesis, including co-clustering with validated activating and recurrent mutations in multiple cancer types (Fig. 3) and are detected in patient tumors

(Fig. 6). Co-clustering events may represent potential drivers and therapeutic targets.

Proteomics datasets, such as those generated by CPTAC, are quantifying increasingly larger numbers of phosphoproteomes in cancer and other samples. The abundant phosphosites of unknown significance (PUS) discovered in these datasets highlight the urgent need for enhanced annotation and prioritization using approaches like HotPho. There are still significant limitations to identifying functional hybrid clusters, as prioritization necessarily relies on known mutations or functional domains. Thus, while we also discovered phosphosite-only clusters, we cannot yet effectively determine their significance until we enhance our understanding of functional phosphosites.

Our investigation supports the functional relevance of co-clustering phosphosites and mutations. For example, we found that these phosphosites and mutations are enriched in functional domains of kinases and histones, that co-clustering mutations tend to be functionally active and confer genetic dependency. Among the 1255 co-clustering phosphosites, only 25 were previously known to be associated with cancer (Supplementary Data 12). Hopefully, the repertoire of characterized phosphosites will grow rapidly using combinations of high throughput proteomics approaches, systematic in silico analysis, and experimental validation[36,37].

More, multiple co-clustering phosphosites were located in activation loops of kinase proteins, including RET p.S904, MET p.Y1248, AKT1 p.T308, EGFR p.Y827/869, as well as the MAPK3-regulated site MAPK8 p.Y185. Crystal structures revealed that PTPN12 p.S275 is found in the Q loop that constitutes the pY+1 pocket demonstrating strong substrate specificity, and the phospho-inhibitory mutant p.S275A significantly decreased the activity of PTPN12 toward all three HER2 phospho-peptides[38]. Notably, this site also harbored a rare mutation p.S275C in the TCGA MC3 mutation data, which only showed clusters when leveraging the adjacent phosphosite information but was not found in the mutation-only clusters.

Spatially co-clustering phosphosites and mutations may interact and exhibit further associations in patient samples. Currently, we only observed a handful of examples where a single tumor sample carries both of the co-clustering phosphosites and mutations in existing quantitative phosphoproteomics cohorts, precluding systematic investigations of their correlative relationships (Supplementary Fig. 9). The growing cohort size of CPTAC and other cancer global phosphoproteomic datasets will likely enable us to test the intriguing hypothesis that samples carrying mutations will show disrupted regulation or mutual exclusivity of a co-clustering phosphosite with sufficient statistical power.

In conclusion, we conduct a large-scale spatial clustering analysis between 225,151 phosphosites and 791,637 missense mutations using the HotPho tool. The resulting 474 hybrid clusters help us discover 1255 phosphosites co-clustering with mutations in human cancer, dozens of which are adjacent to activating mutations and verified in patient tumor samples. Our approach nominates phosphosites of likely functional significance for experimental validation and may be expanded to investigate other post-translational modifications, such as acetylation and glycosylation.

## Methods

### Data sources

*Phosphosites data.* We gathered 225,151 human phosphorylation sites from PTMcosmos, following a procedure similar to our recently published study[39]. PTM sites from PTMcosmos were retrieved from UniProt Knowledge Base (UniProtKB) version 2018.01, PhosphoSitePlus (snapshot on the date 2018–02–14), and CPTAC2 MS phosphoproteome data. A PTM site was included if it satisfied either of the following: (1) included in UniProtKB and was reported in at least one publication or by sequence similarity. (2) included in PhosphoSitePlus and was

reported in at least one publication or validated internally by Cell Signaling Technology. (3) included in CPTAC2 experiments and was detected in at least one of the samples. To match phosphosites between multiple databases, we used transvar[40] to map amino acid residues on different protein isoforms to their unique genomic positions.

*Somatic mutation data.* We used somatic mutations from the TCGA cohort as provided by the publicly-available MC3[11] mutation annotation file (MAF) (syn7824274). These mutations were further filtered based on flagged artifacts, hypermutators, and pathology to a driver discovery dataset of 9062 samples with 791,489 missense mutations, as described in the recent PanCanAtlas somatic driver paper[25].

**Somatic mutation data.** We curated experimentally validated mutations identified as neutral or activating from multiple databases and papers, including the Cancer Biomarkers database within the Cancer Genome Interpreter[18], OncoKB[19], KinDriver[20], and ClinVar[41]. We subsequently required an activating mutation to be seen in at least 2 of these sources, collecting a total of 367 activating mutations.

*PDB structures.* We used the GRCh37 assembly and Ensembl release 75 to pre-process residue pair data for all human proteins in RCSB PDB as of 22 May 2017, which includes PDB structures of 6002 genes.

Some chains or structures from PDB were filtered out due to the following types of artifacts in the data file annotations, (1) chains with inconsistent PDB to UniProt coordinate ranges from DBREF given any alterations from SEQADV length changes, (2) chains where SEQADV describes REMARK 999, which indicates absent residues explained by free text, and (3) structures in which site mismatches were identified (for example, where a Threonine should be found according to the UniProt sequence, but a Valine was instead found at the position in the PDB sequence) even after converting between the PDB and UniProt coordinates designated by the DBREF line.

### Bioinformatics and experimental analyses

*Quality control of sites and structures.* HotPho reads a site input file where each phosphosite must contain the HUGO symbol, Ensembl transcript accession ID (ENST), its residue position within the given transcript, and feature description. The sites are then combined and run through the HotSpot3D search step to produce pairwise data between mutations and phosphosites, comprising mutation–mutation pairs, mutation-site pairs, and site-site pairs. Even though HotPho calculates offsets in residue numbers in PDB structures and transcripts, some offsets provided by structure uploaders resulted in the erroneous mapping of residues. In the resulting pairwise files with phosphosites (.musite and.sites files), we, therefore, filtered out the sites where the mapped residue on the PDB structure differs from those documented in our original input phosphosite file, ultimately retaining 785,867 mutation–mutation pairs, 376,614 mutation-site pairs (78,477 eliminated), and 1,010,011 site-site pairs (267,547 eliminated).

*The HotPho algorithm and cluster discovery.* HotPho extends beyond the originating HotSpot3D algorithm[8] and enables co-clusterings of mutations and phosphosites on protein structures (Fig. 1). Briefly, 3D distances of all missense mutations and phosphosites were calculated using PDB structures, considering the closest atoms on their respective amino acids on PDB structures, to identify proximity pairs. Each potential cluster is then treated as an undirected graph $G = (V,E)$, where $V$ is a subset of the input mutations and phosphosites and $E$ is the set of proximal pairs from $V$. Considering $v_i, v_j \in V$ for $i, j \in \{1,2,...,N\}$ where $N$ is the number of vertices in $V$, the clusters are built up using the Floyd–Warshall shortest-paths algorithm, initiated by the distance matrix of the edges, to obtain the geodesics, $g_{i,j}$ between each $v_i$ and $v_j$. For each $v_i \in V$ where $i \neq j$, the cluster centrality, $c(v_i)$, is then calculated as:

$$c(v_i) = \sum_{j=1}^{N} \frac{1}{2^{g_{i,j}}} \quad (1)$$

For each of the cluster, the centroid is identified as the vertex showing the highest $c(v_i)$, and the cluster closeness score (Cc) is calculated as:

$$Cc = \sum_{i=1}^{N} c(v_i) \quad (2)$$

A high Cc score indicates a dense 3D cluster enriched in recurrent mutations and phosphosites on the protein structure. In the pan-cancer study of mutation-only clusters, clusters with known cancer proteins showed significantly higher Cc score than those without cancer-related proteins, and a threshold at top 5% showed a notably significant difference between cancer- and non-cancer-related proteins[8]. Here in our hybrid cluster analysis, we not only show the top 5% clusters show sensitivity in distinguishing cancer driver genes vs. other genes, but also in observed vs. randomly simulated clusters.

We then conducted clustering using HotPho with recurrence as the vertex type. The analysis generated a total of 30,131 unfiltered clusters in 4989 unique genes, comprising 9483 hybrid clusters, 18,112 mutation-only clusters, and 2536 site-only clusters. To resolve the multi-modal distribution of cluster closeness scores, we

further compared the score distributions for 299 mutation-enriched cancer driver genes[13] versus other genes using a ROC curve analysis (Supplementary Fig. 1B, C).

*Cluster benchmarking using permutation analysis.* The hybrid clusters generated by HotPho was benchmarked by comparison to those obtained through HotPho analyses using a combination of the TCGA MC3 mutation data and permutated phosphosite data. Since we are interested in the hybrid clusters having high cluster closeness (Cc) scores (ie. more closely packed clusters) we chose the top 100 genes having high CC clusters. Next, for each of these genes, we found the number of phosphosites in the original dataset, which are covered by at least one structure. After that, we generated a permutated phosphosites-dataset by randomly populating the sites at possible covered phosphosite residue locations keeping the original residue ratios the same. HotPho clustering was performed for 100 such simulated phosphosites-datasets and the maintained TCGA MC3 mutation call backbone given the non-random distribution and occurrence count of mutation calls. Finally, the clusters from the original HotPho run and the simulated runs were compared focusing on the number of clusters and their Cc score distribution. We further evaluated the sensitivity, specificity, and ROC curves using different threshold of the Cc score (Supplementary Fig. 1D). Based on our simulated results, we set the cluster closeness thresholds as the top 5% cluster closeness score within each cluster type (e.g., hybrid, mutation-only, and phosphosite-only).

**Domain enrichment analysis**. We conducted a domain enrichment analysis of co-cluster phosphosites in PFAM domains (Pfam 31.0 released March 2017)[42]. We evaluated domain enrichment of co-clustered phosphosites using the Fisher test. Each $2 \times 2$ table was comprised of tallies of domain status (current domain or not) versus co-clustered status (co-cluster with mutation or not). Although this test is exact, we followed the general rule-of-thumb for table testing of only evaluating those cases where there were at least 5 mutations and phosphosites in the domain. Resulting *P*-values were corrected to FDR values using the standard Benjamini-Hochberg procedure.

**Mutational impact RPPA analysis**. Similar to our previous analyses of a different set of mutations[27], TCGA level-3 normalized RPPA expression data of the tumor samples were downloaded from Firehose (2016/1/28 archive). The protein/phosphoprotein expression percentile of individual proteins in each cancer cohort was calculated using the empirical cumulative distribution function (ecdf), as implemented in R. Where there are at least 3 carriers within each cancer type, we then applied the linear model to evaluate the protein/phosphoprotein expression percentile between carriers of co-clustered mutations and all other cancer cases. The age at initial diagnosis, gender, and ethnicity are included as covariates to account for potential confounding effects. The resulting *P* values were adjusted to FDR using the standard Benjamini-Hochberg procedure for tests across all cancer types.

**Mutational impact proteome analysis**. We analyzed the effects of clustered mutations using samples from the CPTAC2 retrospective[3,4,43] and prospective collection (https://cptac-data-portal.georgetown.edu/cptac/public). For each hybrid cluster, protein levels were compared between samples with and without cluster mutations (Wilcoxon rank-sum test).

**Cancer cell dependency analyses**. Within each lineage, we carried out a Wilcoxon rank-sum test between the cell lines with co-clustered missense mutations versus those with either (1) other missense mutations, or (2) other non-synonymous mutations. The *p*-values represent the alternative hypothesis that the dependency score distribution of the cell lines with co-clustered mutations is located left (more dependent, more vulnerable) of that of without co-clustered mutations, and they are multiple-testing adjusted using the BH method for FDR.

**Literature reviews of cancer-associated phosphosites**. First, we confined our search space to 71 cancer genes with hybrid clusters by limiting our search space to 299 cancer driver genes[25]. The abstracts of all publications associated with a phosphosite were then retrieved from Europe PMC using their Digital Object Identifier (DOI) or PubMed identifier (PMID). We determined a paper to be cancer-related if its abstract contained the keyword "tumor" and/or "cancer". We then closely examined whether the exact co-clustering phosphosites identified by HotPho showed any alterations on cancer-related phenotypes in these publications. Additionally, we included all disease-associated sites in PhosphoSitePlus (snapshot on the date 2018–02–14) that were connected to any type of cancer.

**Reporting summary**. Further information on research design is available in the Nature Research Reporting Summary linked to this article.

## Data availability

The TCGA somatic mutation data are available at the Genome Data Commons (GDC) [https://gdc.cancer.gov/about-data/publications/mc3-2017/]. CPTAC data are available at the CPTAC Data Portal [https://cptac-data-portal.georgetown.edu/]. The CPTAC datasets used in this study include the CPTAC2 prospective breast [https://cptac-data-

portal.georgetown.edu/study-summary/S039], ovarian [https://cptac-data-portal.georgetown.edu/study-summary/S038], and colorectal [https://cptac-data-portal.georgetown.edu/study-summary/S037] cancer studies, as well as the CPTAC3 discovery endometrial cancer [https://cptac-data-portal.georgetown.edu/study-summary/S053] and clear cell renal carcinoma [https://cptac-data-portal.georgetown.edu/study-summary/S050] studies. Phosphosite data from PTMCosmos are available at https://ptmcosmos.wustl.edu/.

## Code availability

HotPho is released within the package of HotSpot3D v1.7.0 and updated in later versions [available on GitHub: https://github.com/ding-lab/hotspot3d]. HotPho commands and all subsequent analyses scripts used for this manuscript are publicly available [https://github.com/ding-lab/HotPho_Analysis]. Analyses were conducted based on scripts written using the R programming language version 3.3.2 (2016–10–31).

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

## Acknowledgements

This work was supported by the National Cancer Institute grants U24CA160035 to R.T. and M.E., U24CA210972 to D.F., L.D., and S.H.P., U24CA211006 to L.D. and R.G., and National Human Genome Research Institute grant U01HG006517 to L.D. and F.C. We thank The Cancer Genome Atlas (cancergenome.nih.gov) and CPTAC as the source of primary data, and members of the CPTAC Research Network for helpful discussions.

## Author contributions

L.D., K.H., and A.D.S. conceived the idea and designed the HotPho workflow. A.D.S. and A.W. developed the HotPho algorithm and conducted the simulation analysis. K.H. and L.D. designed analyses plans. K.H., D.C.Z., L.W., A.E., R.L., Y.W., J.B., S.S., A.D.S., and A.W. analyzed the data. K.H., D.C., M.A.W., C.L., and R.L. created figures. K.H. wrote the manuscript. L.D., K.C., M.C.W., K.R., S.H.P., B.R., D.F., and G.M. edited the manuscript. G.M. provided experimental validation data. L.D. supervised research. All authors reviewed and approved the manuscript.

## Competing interests

The authors declare no competing interests.
