## [Peer Review File · Nature Communications]

REVIEWER COMMENTS

Reviewer #2 (Remarks to the Author): Expert in computational biology

In this manuscript, the authors report a study on the "clustering" tendency of protein phosphorylation sites and somatic mutations commonly observed in cancers, in 3D structures of proteins. They have compiled a compendium of cluster sites in which either mutations, or phosphorylation sites, or hybrid of both are enriched. The authors discussed characteristics of clusters from multiple perspectives and provided some evidence regarding potential interaction between mutations and phosphorylation sites. Overall, this study provides a useful resource and a hypothesis-generating tool for other researchers to study how mutation events may affect the function of proteins in oncogenesis, whose function may be normally regulated through phosphorylation.

Major comments:

1. The algorithm of HotPho is not clearly described in Methods nor in main text (Section "Cluster discovery using HotPho"). It is unclear how the algorithm defines a cluster and then how to calculate the "closeness score" of a cluster (there is no mathematical definition of "closeness scores"). Fig 1B shows that there is a significant portion of random clusters pass the threshold of "top 5% percentile", and authors did not comment the ratio of the number of clusters (area under the curve) from true data over that of random clusters in the tails beyond the 5% cutoff threshold. This would be critical for a reader to assess the reported results.

Minor comment:

1. It is interesting to see that the "closeness scores" are distributed in a bimodal fashion (Fig 1B left panel), which authors did not explain their understanding. A better understanding of such a distribution pattern may be important for interpreting whether the thresholding used in the manuscript is valid.
2. It would be more informative to add the statistics of "pan-cancer" mutation events in the hybrid clusters presented in the figure. Current presentation suggests different cancer types are enriched with different types of mutations in different clusters, which may not necessarily be true. Statistics from pan-cancer would reflect how often each cluster is likely involved in oncogenic process in all cancers, and cancer-type-specific and cluster-specific statistics should be normalized with respect to the overall mutation frequency of mutation events in a cluster.
3. It is interesting to note that "functional scores" of direct clusters is lower than that of the hybrid clusters. Thoughts from authors?

Reviewer #3 (Remarks to the Author): Expert in signalling

Summary

This paper reviews cancer mutation databases, protein phosphorylation databases, and protein structure databases and reports that mutations and phosphorylations tend to cluster in proximity to each other consistent with the functional significance of these structural regions. This composite

analysis algorithm is used to interrogate TCGA, CPTAC, and other databases. They evaluate several existing mutation assessing algorithms and show that co-clustering mutations identified by their algorithm have higher predicted functional scores. A database of experimental data is also mined to show that what they describe as co-clustering mutations are more likely to be functional in cell-based assays than other mutations.

Review

This manuscript is generally well written. The figures are well prepared and clear. The data is interesting and their algorithm would be of interest and a welcome addition to the field. The only thing is that the novelty and impact of this work is modest. The idea that cancer mutations and protein phosphorylation are functionally related and the idea that mutations cluster at functionally important regions are well described phenomenon and many of these reports are cited in this manuscript. 3D clustering of mutations is well described (PMID 28115009, 19834613, 22252508, 26485003, 27043210, 26392535, 27150811). The fact that mutations affect phosphorylation and signaling is well described (PMID 23340843, 24089029, 28170390). Others have described mutational clustering according to signaling pathways and protein networks (PMID 28170390, 30297789). The algorithm presented in this paper is perhaps novel and interesting, and their designation of the “co-clustered” mutations is perhaps a new designation. But the ultimate contribution and impact of this paper is incremental in nature. Although this work is not novel in concept, it expands the tools and techniques for prioritizing mutations and perhaps expands the catalogue of suspected driver mutations and functionally important phosphorylations. As an example of the incremental nature of the impact, the abstract highlights BRAF T599 and S602 as a co-clustering phosphosite. Is this really a novel contribution? The fact that these are in proximity to the hot spot residue V600 and important in regulating the activation loop of BRAF is well known (PMID 26657898, 11032810). Similarly the functional significance of the RB1 phosphorylation highlighted in the abstract has been extensively characterized (PMID 10499802, 20871633). The contribution of this work is probably more in validating current concepts and only incremental in proposing new ones. The development and announcement of a new computation tool is also a contribution.

Specific critiques

The section regarding the kinase regulation of clustering phosphosites is difficult to understand. I have read it again and again and I can't really understand what is being described or suggested or what is the important outcome of this. There is repeated reference to “regulated phosphosites”. So is the assertion that if a phosphosite-to-kinase relationship is known in the literature, then this is an important regulatory phosphorylation and if it is not described, then these are random phosphorylations on proteins that are of no functional significance? Are you assuming an entity of “passenger phosphorylations” the same way we think there are passenger mutations? Every phosphorylation must involve a kinase, whether or not it has been identified in the literature. And to assume that if it has not been described, then it must not be important seems naïve. I don't really understand the contribution of this section of the manuscript. Perhaps it is not written in a manner that is comprehensible to a broad readership.

Reviewer #4 (Remarks to the Author): expert in proteomics

Li Ding and co-workers describe a method to identify candidates for cancer-driving protein phosphorylation sites out of the vast amount of phosphorylation sites accessible through a growing body of mass spec-generated phosphoproteomics dataset. The analysis is based on an algorithm incorporating protein structure into the determination of clusters of amino acid positions of interest. The group has previously used this analysis strategy to define clusters of mutations in cancer (PMID: 27294619). They are now extending the approach by not only including mutations but also phosphosites, and they are explicitly focusing on co-clusters of phosphosites and mutations as they may reveal the annotated phosphosites to have a cancer-driving function. I think this is an interesting approach, and it is promising in helping us to bring some direction into mining the vast and understudied data from phosphoproteomics experiments. Some experimental follow-up would really boost the study's impact; I will get back to that later. One major issue for me is that although the novelty aspect of the paper is on focusing on including spatially interacting phosphosites and mutations but that all of the validations of the identified clusters do not distinguish between sites in close linear proximity to the sites. Throughout the manuscript, I was wondering how much new information was revealed through sites identified just based on proximity driven by protein structure rather than through linear proximity. The manuscript is also a little lengthy, and I wonder if seven figures are required to describe the study. Further, detailed comments are listed below. I would like to ask the authors to address these comments before I would considering recommending the publication of the study in Nature Communications.

(1) Figure 1B: I am surprised how little of an effect the randomization of phosphosites has on the clustering. I would like to see the clustering upon the randomization of the mutations. Is it that all the clustering we see is mainly driven by the mutations, and randomly distributed phosphosites are sometimes co-localized with the mutation clusters? The co-clustering of which phosphosites may still be of high interest; however, some of the follow-up validations of finding cancer-relevance for the phospho-mutation co-clusters may be less relevant if the cancer-relevance is only driven by the mutation clusters. Minor point: Why do we see a bimodal distribution for the cluster-closeness histogram? Is this showing a clear distinguishing between clusters and non-clusters? If so, why do we see a less pronounced bimodal distribution for the real data over the randomized data?

(2) How many of the 474 hybrid clusters would have been identified as mutation-only clusters if the phosphosites were excluded from the analysis? This goes back to the point (1) and to the question if phosphosites might be merely a decoration to mutation clusters.

(3) I think Figures 3 and 4 could be combined into one figure with putting some clusters shown on 3D protein structures into Supp Figures.

(4) Figure 5. I think this is the most interesting aspect of the study. Fig. 5E: It is not clear what phosphoform of the kinases is taken for the correlation. I think a phosphorylation event at the activation loop would be most useful. I would also suggest extending the model for building clusters by including mutations and phosphosites on the kinases likely catalyzing the clustering phosphosites.

(5) Figure 7. We are looking at very prominent mutation clusters that are found to be co-clustering with phosphosites. I wonder if there are other properties of these clusters (besides the co-clustering phosphosites) that may explain the higher functionality of the co-clustering mutations. Are there more mutations found in these clusters over the background? Are these mutations found at elevated frequency in cancer? I would suggest another validation focusing on the kinases linked to the co-clustered phosphosites. If the sites are functionally important, depleting the kinases should show an effect. A good start would be to work with existing genetic drop-put screen data (PMID: 28753430) and validate some kinases in cell lines using RNAi.

Authors: We appreciate the positive comments and constructive feedback from the three reviewers in the last round of review. Following the recommendations, the manuscript has been revised. Please see below a point-by-point response, including multiple changes in three major categories:

1. **Benchmarking of the cluster closeness scores**, including the mathematical definition of the score, receiver operating characteristic (ROC) analyses of observed vs. simulated clusters, and sensitivity to distinguish cancer driver vs. other genes.
2. **Additional analyses demonstrating the value of phosphosite co-clustering**, including cancer-specificity of co-clustering mutations, comparing HotPho results to a mutation-only run, demonstrating co-clustering status distinguishes functional mutations beyond mutation recurrence, and dependency analyses of kinases with co-clustering substrate phosphosites.
3. **Manuscript edits**, including highlighting novel relationships uniquely found using 3D methods and phosphosites in activation loops, merging/improving Figure 3/4, and multiple tracked edits to enhance clarity of the original contributions.

We believe the revised manuscript adequately addresses all concerns and hope you will find it to be satisfactory.

REVIEWER COMMENTS

Reviewer #2 (Remarks to the Author): Expert in computational biology

In this manuscript, the authors report a study on the "clustering" tendency of protein phosphorylation sites and somatic mutations commonly observed in cancers, in 3D structures of proteins. They have compiled a compendium of cluster sites in which either mutations, or phosphorylation sites, or hybrid of both are enriched. The authors discussed characteristics of clusters from multiple perspectives and provided some evidence regarding potential interaction between mutations and phosphorylation sites. Overall, this study provides a useful resource and a hypothesis-generating tool for other researchers to study how mutation events may affect the function of proteins in oncogenesis, whose function may be normally regulated through phosphorylation.

Authors: We appreciate the Reviewer's positive assessments of the useful resource and tool provided by our manuscript.

Major comments:

1. The algorithm of HotPho is not clearly described in Methods nor in main text (Section "Cluster discovery using HotPho"). It is unclear how the algorithm defines a cluster and then how to calculate the "closeness score" of a cluster (there is no mathematical definition of "closeness scores"). Fig 1B shows that there is a significant portion of random clusters pass the threshold of "top 5% percentile", and authors did not comment the ratio of the number of clusters (area under the curve) from true data over that of

random clusters in the tails beyond the 5% cutoff threshold. This would be critical for a reader to assess the reported results.

Authors: We thank the reviewer for the comment and included more details on how the cluster closeness score is calculated in the **Methods**,

“The HotPho algorithm and cluster discovery

HotPho extends beyond the originating HotSpot3D algorithm⁸ and enables co-clustering of mutations and phosphosites on protein structures (**Figure 1**). Briefly, 3D distances of all missense mutations and phosphosites were calculated using PDB structures, considering the closest atoms on their respective amino acids on PDB structures, to identify proximity pairs. Each potential cluster is then treated as an undirected graph $G = (V,E)$, where V is a subset of the input mutations and phosphosites and E is the set of proximal pairs from V . Considering $v_i, v_j \in V$ for $i, j \in \{1,2,\dots,N\}$ where N is the number of vertices in V , the clusters are built up using the Floyd–Warshall shortest-paths algorithm, initiated by the distance matrix of the edges, to obtain the geodesics, $g_{i,j}$ between each v_i and v_j . For each $v_i \in V$ where $i \neq j$, the cluster centrality, $c(v_i)$, is then calculated as:

$$c(v_i) = \sum_{j=1}^N \frac{1}{2^{g_{i,j}}}$$

For each of the cluster, the centroid is identified as the vertex showing the highest $c(v_i)$, and the closeness score (C_c) is calculated as:

$$C_c = \sum_{i=1}^N c(v_i)$$

A high C_c score indicates a dense 3D cluster enriched in recurrent mutations and phosphosites on the protein structure. Further benchmarking of the C_c score and further details of the HotSpot3D algorithm are as previously described⁸.

We also conducted further analyses to investigate the cluster closeness score distribution based on receiver operating characteristic (ROC) curve analysis of observed vs. simulated clusters as well as score distribution analysis of driver gene vs. non-driver gene clusters. The Results are now added to the **Results**,

“We defined the criteria of high-confidence clusters to have cluster closeness scores within the top 5% of their respective cluster types and subsequently limited our analyses to these clusters. In hybrid clusters, the 5% sensitivity corresponded to 97.4% specificity in a receiver operating characteristic (ROC) curve analysis (AUC = 0.58, **Supplementary Fig. 1A**). Further, while this threshold (cluster closeness score = 2.56) may permit false-positives if the simulated phosphosites only contain negatives, we observed many of the clusters containing activating or recurrent mutations with cluster closeness scores close to the threshold (Supplementary Table 3). It is possible that the spatial distribution of cancer mutations and commonality phosphosite residues

(i.e., serine, threonine, and tyrosine) is not random and thus retaining these additional hybrid clusters is needed to minimize false-negatives. Lastly, to resolve possible reasons underlying the multi-modal distribution of cluster closeness scores, we compared the score distributions for 299 mutation-enriched cancer driver genes versus other genes. While hybrid clusters involving driver genes showed a higher density at the higher-score mode, driver gene status did not guarantee high scores (**Supplementary Fig. 1B**). The 5% score threshold showed a sensitivity = 0.17 and specificity = 96.0% in distinguishing hybrid clusters with driver genes (**Supplementary Fig. 1C**). Cluster closeness scores for all identified clusters are available to prioritize a more stringent set of clusters (**Supplementary Table 1**)."

Minor comment:

1. It is interesting to see that the "closeness scores" are distributed in a bimodal fashion (Fig 1B left panel), which authors did not explain their understanding. A better understanding of such a distribution pattern may be import for interpreting whether the thresholding used in the manuscript is valid.

Authors: Please see response to major comment above.

2. It would be more informative to add the statistics of "pan-cancer" mutation events in the hybrid clusters presented in the figure. Current presentation suggests different cancer types are enriched with different types of mutations in different clusters, which may not necessarily be true. Statistics from pan-cancer would reflect how often each cluster is likely involved in oncogenic process in all cancers, and cancer-type-specific and cluster-specific statistics should be normalized with respect to the overall mutation frequency of mutation events in a cluster.

Authors: We have now revised **Figure 3A** using the size of the dot to indicate the pan-cancer score calculated by the averaging the frequencies of the mutation across the 32 cancer types in TCGA. The co-clustered mutations with the highest pan-cancer score are IDH1 p.R132H (0.025), KRAS p.G12D (0.024), KRAS p.G12V (0.022), and PIK3CA p.H1047R (0.016), and their co-clustered phosphosites are as shown in the Figure.

3. It is interesting to note that "functional scores" of direct clusters is lower than that of the hybrid clusters. Thoughts from authors?

Authors: We agree that the observation where the functional scores being the highest for co-clustered mutations, compared to each of the "Direct", "Proximal", or "None" mutations is interesting (each test had Wilcoxon rank-sum test $P < 2.2e-16$), albeit the score distributions appear more similar between Direct, Proximal, Clustered compared to None. One possibility is that the HotPho clustering require recurrence weighing, and thus, the clustered sites are more likely to include those near more recurrent mutations compared to the Direct category, where we considered any mutation that overlapped phosphosites. The other possibility is that activating mutations with high scores may bias for those adjacent to crucial phosphosites instead of directly locating on the

phosphosite. A myriad of mechanisms affect which residues get mutated. From this analysis, we do not attempt to overdraw conclusions but focused on demonstrating the effectiveness of identifying functional residues by 3D clustering of mutations and phosphosites.

Reviewer #3 (Remarks to the Author): Expert in signalling

Summary

This paper reviews cancer mutation databases, protein phosphorylation databases, and protein structure databases and reports that mutations and phosphorylations tend to cluster in proximity to each other consistent with the functional significance of these structural regions. This composite analysis algorithm is used to interrogate TCGA, CPTAC, and other databases. They evaluate several existing mutation assessing algorithms and show that co-clustering mutations identified by their algorithm have higher predicted functional scores. A database of experimental data is also mined to show that what they describe as co-clustering mutations are more likely to be functional in cell-based assays than other mutations.

Review

This manuscript is generally well written. The figures are well prepared and clear. The data is interesting and their algorithm would be of interest and a welcome addition to the field. The only thing is that the novelty and impact of this work is modest. The idea that cancer mutations and protein phosphorylation are functionally related and the idea that mutations cluster at functionally important regions are well described phenomenon and many of these reports are cited in this manuscript. 3D clustering of mutations is well described (PMID 28115009, 19834613, 22252508, 26485003, 27043210, 26392535, 27150811). The fact that mutations affect phosphorylation and signaling is well described (PMID 23340843, 24089029, 28170390). Others have described mutational clustering according to signaling pathways and protein networks (PMID 28170390, 30297789). The algorithm presented in this paper is perhaps novel and interesting, and their designation of the “co-clustered” mutations is perhaps a new designation. But the ultimate contribution and impact of this paper is incremental in nature. Although this work is not novel in concept, it expands the tools and techniques for prioritizing mutations and perhaps expands the catalogue of suspected driver mutations and functionally important phosphorylations. As an example of the incremental nature of the impact, the abstract highlights BRAF T599 and S602 as a co-clustering phosphosite. Is this really a novel contribution? The fact that these are in proximity to the hot spot residue V600 and important in regulating the activation loop of BRAF is well known (PMID 26657898, 11032810). Similarly the functional significance of the RB1 phosphorylation highlighted in the abstract has been extensively characterized (PMID 10499802, 20871633). The contribution of this work is probably more in validating current concepts and only incremental in proposing new ones. The development and announcement of a new computation tool is also a contribution.

Authors: We thank the Reviewer’s positive comments on the prose, figure, and computational tool presented by the manuscript. The new computational tool, HotPho,

nominated over 1,000 phosphosites which provided the readers a comprehensive list that may validate existing findings, build evidence, or prioritize new sites. We note that conceptually, any phosphosites with antibodies probably had a mention in the literature. HotPho analyses reveal extensive numbers of sites that we provided the readers in the Supplementary Table, but given the space and lack of further evidence, the novel findings with little other evidence were not extensively described in main text.

To highlight our unique contributions, we now describe HotPho-unique events not found by linear methods in the **Abstract**,

“HotPho identified 474 such hybrid clusters containing a total of 1,255 co-clustering phosphosites, including RET p.S904/Y928, the conserved HRAS/KRAS p.Y96, and IDH1 p.Y139/IDH2 p.Y179 adjacent to known activating or recurrent mutations on proteins yet not found in linear proximity.”

This is aside from other sites verified/previously implicated in cancer for which HotPho also added strong supporting evidence (CTNNB1 p.S29/Y30, EGFR p.S720, MAPK1 p.S142, and PLG p.S358, and PTPN12 p.S275). In instances where we have not cited them, we have added the reference the reviewer suggested to the last paragraph in **Results** discussing literature evidence of the identified phosphosites. Rather than suggesting the 1,000 phosphosites did not identify any new events, the discussion of these known phosphosites anchor the “positive controls” HotPho was able to prioritize. We now end the paragraph stating,

“These findings further validate the functionality of selected co-clustering phosphosites HotPho identified in hybrid clusters and suggest other sites in hybrid clusters may be prioritized for downstream investigations.”

Finally, we also added a paragraph highlighting phosphosites in kinase activation loops to the **Discussion**,

“More, multiple co-clustering phosphosites were located in activation loops of kinase proteins, including RET p.S904, MET p.Y1248, AKT1 p.T308, EGFR p.Y827/869, as well as the MAPK3-regulated site MAPK8 p.Y185. Crystal structures revealed that PTPN12 p.S275 is found in the Q loop that constitutes the pY+1 pocket demonstrating strong substrate specificity, and the phospho-inhibitory mutant p.S275A significantly decreased the activity of PTPN12 toward all three HER2 phospho-peptides⁴¹. Notably, this site also harbored a rare mutation p.S275C in the TCGA MC3 mutation data, which only showed clusters when leveraging the adjacent phosphosite information but was not found in the mutation-only clusters.”

Specific critiques

The section regarding the kinase regulation of clustering phosphosites is difficult to understand. I have read it again and again and I can't really understand what is being described or suggested or what is the important outcome of this. There is repeated reference to “regulated phosphosites”. So is the assertion that if a phosphosite-to-kinase relationship is known in the literature, then this is an important regulatory phosphorylation and if it is not described, then these are random phosphorylations on

proteins that are of no functional significance? Are you assuming an entity of “passenger phosphorylations” the same way we think there are passenger mutations? Every phosphorylation must involve a kinase, whether or not it has been identified in the literature. And to assume that if it has not been described, then it must not be important seems naïve. I don’t really understand the contribution of this section of the manuscript. Perhaps it is not written in a manner that is comprehensible to a broad readership.

Authors: We thank the reviewer for the feedback and agree that all phosphosites are regulated by kinases. In this section, we have now revised the term “regulated phosphosites” to “phosphosites with known regulations”.

Reviewer #4 (Remarks to the Author): expert in proteomics

Li Ding and co-workers describe a method to identify candidates for cancer-driving protein phosphorylation sites out of the vast amount of phosphorylation sites accessible through a growing body of mass spec-generated phosphoproteomics dataset. The analysis is based on an algorithm incorporating protein structure into the determination of clusters of amino acid positions of interest. The group has previously used this analysis strategy to define clusters of mutations in cancer (PMID: 27294619). They are now extending the approach by not only including mutations but also phosphosites, and they are explicitly focusing on co-clusters of phosphosites and mutations as they may reveal the annotated phosphosites to have a cancer-driving function. I think this is an interesting approach, and it is promising in helping us to bring some direction into mining the vast and understudied data from phosphoproteomics experiments. Some experimental follow-up would really boost the study's impact; I will get back to that later. One major issue for me is that although the novelty aspect of the paper is on focusing on including spatially interacting phosphosites and mutations but that all of the validations of the identified clusters do not distinguish between sites in close linear proximity to the sites. Throughout the manuscript, I was wondering how much new information was revealed through sites identified just based on proximity driven by protein structure rather than through linear proximity. The manuscript is also a little lengthy, and I wonder if seven figures are required to describe the study. Further, detailed comments are listed below. I would like to ask the authors to address these comments before I would considering recommending the publication of the study in Nature Communications.

Authors: We thank the Reviewer for highlighting the promise of the approach, and agree clarifying the phosphosites HotPho identified beyond linear methods will be useful. We now highlight the significance of the novel phosphosites that are uniquely identified by 3D approach, including revising the sentence in the **Abstract**,

“HotPho identified 474 such hybrid clusters containing a total of 1,255 co-clustering phosphosites, including RET p.S904/Y928, the conserved HRAS/KRAS p.Y96, and IDH1 p.Y139/IDH2 p.Y179 adjacent to known activating or recurrent mutations on proteins yet not found in linear proximity.”

Further, in the Supplementary Table 1 containing all the identified, co-clustering mutations and phosphosites in hybrid clusters, we now include a SiteType column which indicates residues that are Direct/Proximal/Clustered. In addition to the **Results** that already describes the numbers of new sites, we now referred again to this Table for the readers' discretion.

“Among the 1,255 co-clustered phosphosites, 291 sites directly overlap and 356 sites are proximal (within 2 amino acid residues linearly) to their co-clustered mutations (**Supplementary Table 1, Figure 2B**). The HotPho co-clustering analysis adds a substantial count of 608 phosphosites which are distant in terms of a linear sequence, yet close in 3D protein structure, including the majority of the sites found on ACTB, HIST1H2BC, and ERBB2. Nearly half of the clusters we identified can only be found by integrating 3D protein structure, demonstrating the added value of 3D approaches for the discovery of spatial relationships between mutations and phosphosites.”

Additional comments addressing the reviewer's other feedback are described below.

(1) Figure 1B: I am surprised how little of an effect the randomization of phosphosites has on the clustering. I would like to see the clustering upon the randomization of the mutations. Is it that all the clustering we see is mainly driven by the mutations, and randomly distributed phosphosites are sometimes co-localized with the mutation clusters? The co-clustering of which phosphosites may still be of high interest; however, some of the follow-up validations of finding cancer-relevance for the phospho-mutation co-clusters may be less relevant if the cancer-relevance is only driven by the mutation clusters. Minor point: Why do we see a bimodal distribution for the cluster-closeness histogram? Is this showing a clear distinguishing between clusters and non-clusters? If so, why do we see a less pronounced bimodal distribution for the real data over the randomized data?

Authors: We thank the reviewer for the comment. Indeed, the cluster closeness score distributions of observed vs. simulated clusters had overlaps, and thus we picked a stringent 5% threshold within the observed scores to permit very little fraction of false positives. This is evident from our further analyses that investigated the cluster closeness score distribution based on receiver operating characteristic (ROC) curve analysis of observed vs. simulated clusters as well as score distribution analysis of driver gene vs. non-driver gene clusters. These are now added to the **Results**,

“We defined the criteria of high-confidence clusters to have cluster closeness scores within the top 5% of their respective cluster types and subsequently limited our analyses to these clusters. In hybrid clusters, the 5% sensitivity corresponded to 97.4% specificity in a receiver operating characteristic (ROC) curve analysis (AUC = 0.58, **Supplementary Fig. 1A**). Further, while this threshold (cluster closeness score = 2.56) may permit false-positives if the simulated phosphosites only contain negatives, we observed many of the clusters containing activating or recurrent mutations with cluster closeness scores close to the threshold (Supplementary Table 3). It is possible that the spatial distribution of cancer mutations and commonality phosphosite residues (i.e., serine, threonine, and tyrosine) is not random and thus retaining these

additional hybrid clusters is needed to minimize false-negatives. Lastly, to resolve possible reasons underlying the multi-modal distribution of cluster closeness scores, we compared the score distributions for 299 mutation-enriched cancer driver genes versus other genes. While hybrid clusters involving driver genes showed a higher density at the higher-score mode, driver gene status did not guarantee high scores (**Supplementary Fig. 1B**). The 5% score threshold showed a sensitivity = 0.17 and specificity = 96.0% in distinguishing hybrid clusters with driver genes (**Supplementary Fig. 1C**). Cluster closeness scores for all identified clusters are available to prioritize a more stringent set of clusters (**Supplementary Table 1**).”

We also note permuting mutations using the HotSpot3D tool had a large parameter space, such as (1) whether to keep the mutated/new residue fraction and their corresponding relationships consistent, (2) whether to keep the mutations within one domain/gene or across domain/genes (ex. what would be the effect of spreading out high counts of TP53 mutations?), (3) whether to keep mutations within or across cancer types, etc. Such analyses needed to be comprehensively examined by extensive analyses and may reveal new insights on mutation-only clusters and their relationship with recurrence. Given this study focuses on phosphosites co-clustering with mutations instead of mutation-only clusters, we believe the aforementioned task is out of scope of this manuscript.

(2) How many of the 474 hybrid clusters would have been identified as mutation-only clusters if the phosphosites were excluded from the analysis? This goes back to the point (1) and to the question if phosphosites might be merely a decoration to mutation clusters.

Authors: We now conducted a HotSpot3D run using the exact set of MC3 mutations and same recurrence weighing parameters. The analysis resulted in 1,433 unique clusters passing the 5% cc score threshold, containing 9,403 unique mutations. Among the 2,938 mutations found in the 474 hybrid clusters, we found only 48 mutations not found in these mutation-only clusters. The list of 48 mutations contained new mutations of interest in PDE1B (5 mutations), SRSF7 (4 mutations), and PTPN12 p.S275F/C that co-localized with p.S275 and co-clustered with p.S39/p.T40.

We note that given the ability to prioritize cancer somatic mutations based on their recurrence, the mutations have added clustering weights (recurrence) but not the phosphosites, thus the mutation-only run likely encompass most covered phosphosites. The molecular mechanisms of many recurrent missense mutations are unknown, thus even when phosphosites do not provide crucial weights in this version of clustering analyses, the results may reveal recurrent mutations through which the co-clustering phosphosites manifest their effects. Further, the co-clustering analyses can uniquely prioritize phosphosites.

(3) I think Figures 3 and 4 could be combined into one figure with putting some clusters shown on 3D protein structures into Supp Figures.

Authors: As the reviewer suggested, we have now combined the figures into one main Figure 3, put the other clusters into Supplementary Fig. 4, and update all subsequent figure numbers.

(4) Figure 5. I think this is the most interesting aspect of the study. Fig. 5E: It is not clear what phosphoform of the kinases is taken for the correlation. I think a phosphorylation event at the activation loop would be most useful. I would also suggest extending the model for building clusters by including mutations and phosphosites on the kinases likely catalyzing the clustering phosphosites.

Authors: We thank the reviewer for praising the results presented in this section. We now added in the Methods that the phosphoprotein level is based on the average of all observed phosphosites on that given kinase. We also note the expanded version of this quantitative association analysis is now published²³, and we modified the language throughout to reflect that, i.e., “As previously described²³”.

According to the reviewer’s suggestion, we also identified and discussed multiple co-clustering phosphosites located within the kinase signaling loops in a dedicated section of **Discussion**:

“More, multiple co-clustering phosphosites were located in activation loops of kinase proteins, including RET p.S904, MET p.Y1248, AKT1 p.T308, EGFR p.Y827/869, as well as the MAPK3-regulated site MAPK8 p.Y185. Crystal structures revealed that PTPN12 p.S275 is found in the Q loop that constitutes the pY+1 pocket demonstrating strong substrate specificity, and the phospho-inhibitory mutant p.S275A significantly decreased the activity of PTPN12 toward all three HER2 phospho-peptides⁴¹. Notably, this site also harbored a rare mutation p.S275C in the TCGA MC3 mutation data, which only showed clusters when leveraging the adjacent phosphosite information but not in the mutation-only clusters.”

(5) Figure 7. We are looking at very prominent mutation clusters that are found to be co-clustering with phosphosites. I wonder if there are other properties of these clusters (besides the co-clustering phosphosites) that may explain the higher functionality of the co-clustering mutations. Are there more mutations found in these clusters over the background? Are these mutations found at elevated frequency in cancer? I would suggest another validation focusing on the kinases linked to the co-clustered phosphosites. If the sites are functionally important, depleting the kinases should show an effect. A good start would be to work with existing genetic drop-put screen data (PMID: 28753430) and validate some kinases in cell lines using RNAi.

Authors: We now examined several possibilities of the features distinguishing the activating vs. neutral co-clustering mutations, and have added the results to **Results**,

“To evaluate the added predictive power of mutation functionality provided by phosphosite co-clustering, we examined the relationships between mutation functionality versus co-clustering mutation counts, mutation recurrence, and co-clustering with phosphosites. First, using a multivariate logistic regression model corrected for the mutated gene, we found that the number of co-clustering mutations was not significantly associated with the mutation functionality in either the BAF3 (P = 0.91) or MCF10A (P = 0.40) cell line (**Supplementary Fig 9A**).

Second, using a multivariate logistic regression model corrected for the mutated gene, we found that the recurrence of mutations in the TCGA MC3 cohort was significantly associated with the mutation functionality in both the BAF3 ($P = 2.8e-3$) or MCF10A ($P = 0.023$) cell lines. But, when adding the phosphosite co-clustering status to the regression model, the mutation functionality was no longer associated with recurrence ($P > 0.21$), but strongly associated with the co-clustering status in both BAF3 ($P = 1.4e-10$) and MCF ($P = 1.5e-4$) cell lines (**Supplementary Fig 9B**). Altogether, these results suggest that spatial co-clustering with phosphosites may improve prediction of mutation functionality beyond the commonly used mutation recurrence.”

As the reviewer suggested, we conducted analyses of kinases associated with the co-clustered phosphosites using the DepMap CRISPR knockout screen data. The findings were also included in **Results**,

“Next, we sought to investigate whether kinases with co-clustering substrate phosphosites may show higher dependency by cancer cells. Matching HotPho results to the PhosphositePlus²⁴ kinase-substrate database, we identified 61 kinases with at least one co-clustering substrate phosphosites and 285 other kinases without any co-clustering substrate phosphosites among the 346 unique kinases with records. We then conducted correlation analyses to identify expression-associated dependency in cancer cell lineages characterized by the CRISPR-knockout screens in the Cancer Dependency Map (DepMap) project²⁷. We rationalized that if the kinases are oncogenic, cancer cells of the same lineage would exhibit negative correlations between the CERES dependency score and the gene expression of the kinase—where the cancer cells highly expressing the kinases were more vulnerable when the kinase was knockout (ex. ERBB2 knockout in breast cancer cells). Among the 9 lineages included in the analysis, we detected a significantly lower score for kinases linked to any co-clustering phosphosites in breast cancer cell lines (Two-tailed Kolmogorov–Smirnov test, $p = 0.017$) albeit not in 8 other lineages ($p \geq 0.18$, Supplementary Fig. 6).”

REVIEWER COMMENTS

Reviewer #2 (Remarks to the Author):

The authors have adequately addressed my concerns.

Xinghua Lu

Reviewer #3 (Remarks to the Author):

In my original review I was generally complementary, but had two points to make. The first was that the relationships between mutations, phosphorylations, and structure is well recognized and described, and the contribution of the new algorithm is incremental. In the revised manuscript the authors revised the abstract to highlight some of the non-linear mutations/phosphosites that their algorithm has discovered. That's fine and the editor can judge whether the new findings are significant enough for this journal.

My second point was regarding the section entitled "kinase regulation of co-clustering phosphosites". I fail to understand this section, what it is they are trying to show, what it is they think they have shown, and what it all means. Without explicitly saying so, this section seems to be premised on the assumption that many phosphorylations are biologically relevant and many phosphorylations are biologically irrelevant and you can identify the relevant ones from the literature. I asked for clarification and revision so that I and perhaps most readers can understand this section. In the revision they merely changed the wording from "regulated phosphosites" to "phosphosites with known regulations". This semantic revision doesn't help me understand this section any better. It remains unclear to me, and I think to most readers, what is being advanced in this section. In the absence of a direct response from the authors, I must make my own assessment of what is being pushed in this section. From what I can tell, they are making the claim that their algorithm is so good that it can identify biologically relevant phosphorylations from the irrelevant ones, and furthermore it is so good that it can identify driver kinases based on these phosphorylation patterns. I think there is far too little here to support such claims and at best this effort is exploratory in nature. This work is computational and computational studies often require assumptions to generate hypotheses or conclusions and you can get carried away with these assumptions. For example there is an assumption that the higher the expression of a kinase, the more it phosphorylates its substrates. This may have a few anecdotes such as amplified and massively overexpressed kinases, but as a general assumption it is utterly false. As another example, they feel like these computational studies can identify structurally important phosphorylations and their associated kinases and this identifies so called "higher dependency" kinases that can potentially be targeted to treat cancer. This claim is also fraught with problems. Phosphorylations can promote physical association or they can disrupt physical associations of regions of proteins or among proteins with the latter being more likely. And then that promotion or disruption can have a positive or negative effect on the signaling function of that target protein. And the signaling function of that target protein can be tumor-promoting or tumor-suppressing. There is so much complexity that is neglected in this section that I cannot see any value in it.

Reviewer #4 (Remarks to the Author):

I am on the fence regarding recommending the publication of the revised manuscript.

(i) Showing that mutations co-clustered with phosphorylations are enriched in activating mutations (as described in a response to comment 5 in my previous review) is big and – for me – fulfills the requirement for publication in Nature Communications. I think this is a key result and should be highlighted in the abstract.

(ii) The fact that authors found 48 mutations through co-clustering that were not covered in the mutation-only clustering is interesting and should be included in the manuscript.

(iii) I think that the analysis in combination with the DepMap data is not very clear. Kinase expression levels are not necessarily correlated with an increased kinase activity. For this analysis I rather thought of: (a) determining driver-mutations enriched in cell lines of a cancer type, (b) selecting for genes carrying driver mutations that also show an increased fitness effect for these cell lines, (c) check if kinases associated with co-clustering phosphosites on these driver genes also are showing a significantly enriched essentiality for cell line models of this cancer type.

(iv) My question on the bimodal nature of the distribution shown in Fig. 1B, and why the extent of the bimodality is different between the HotPho data and the simulated data, and how this could be used to filter for interesting co-clusters, has been mainly ignored in the rebuttal although it also was brought up by another reviewer.

(v) Most of the work on addressing suggestions of the reviewers from the first round was mainly added in the form of Supp. Figures. As the (main) figures are the windows to a scientific publication, I am not sure these very useful additions and deeper analyses will benefit the readership in an optimal manner.

I would like the authors to address points (ii) – (v) before fully supporting the publication of the manuscript in Nature Communications.

Authors: We appreciate the positive comments and constructive feedback from the three reviewers in the second round of review. Following the recommendations, we conducted additional analyses and the manuscript has been revised thoroughly in related sections. Please see below a point-by-point response. We believe the revised manuscript adequately addresses all concerns and hope you will find it to be satisfactory for publication.

REVIEWER COMMENTS

Reviewer #2 (Remarks to the Author):

The authors have adequately addressed my concerns.

Xinghua Lu

Authors: We thank the reviewer the constructive review.

Reviewer #3 (Remarks to the Author):

In my original review I was generally complementary, but had two points to make. The first was that the relationships between mutations, phosphorylations, and structure is well recognized and described, and the contribution of the new algorithm is incremental. In the revised manuscript the authors revised the abstract to highlight some of the non-linear mutations/phosphosites that their algorithm has discovered. That's fine and the editor can judge whether the new findings are significant enough for this journal.

My second point was regarding the section entitled "kinase regulation of co-clustering phosphosites". I fail to understand this section, what it is they are trying to show, what it is they think they have shown, and what it all means. Without explicitly saying so, this section seems to be premised on the assumption that many phosphorylations are biologically relevant and many phosphorylations are biologically irrelevant and you can identify the relevant ones from the literature. I asked for clarification and revision so that I and perhaps most readers can understand this section. In the revision they merely changed the wording from "regulated phosphosites" to "phosphosites with known regulations". This semantic revision doesn't help me understand this section any better. It remains unclear to me, and I think to most readers, what is being advanced in this section. In the absence of a direct response from the authors, I must make my own assessment of what is being

pushed in this section. From what I can tell, they are making the claim that their algorithm is so good that it can identify biologically relevant phosphorylations from the irrelevant ones, and furthermore it is so good that it can identify driver kinases based on these phosphorylation patterns. I think there is far too little here to support such claims and at best this effort is exploratory in nature. This work is computational and computational studies often require assumptions to generate hypotheses or conclusions and you can get carried away with these assumptions. For example there is an assumption that the higher the expression of a kinase, the more it phosphorylates its substrates. This may have a few anecdotes such as amplified and massively overexpressed kinases, but as a general assumption it is utterly false. As another

example, they feel like these computational studies can identify structurally important phosphorylations and their associated kinases and this identifies so called “higher dependency” kinases that can potentially be targeted to treat cancer. This claim is also fraught with problems. Phosphorylations can promote physical association or they can disrupt physical associations of regions of proteins or among proteins with the latter being more likely. And then that promotion or disruption can have a positive or negative effect on the signaling function of that target protein. And the signaling function of that target protein can be tumor-promoting or tumor-suppressing. There is so much complexity that is neglected in this section that I cannot see any value in it.

Authors: We agree with the reviewer that compared to other sections of the already-lengthy manuscript, this section of annotating the regulator of the co-clustering phosphosites add less conceptual advances. We thank the reviewer for the thoughtful feedback and have removed the section, as well as other associated texts throughout, from the manuscript.

Reviewer #4 (Remarks to the Author):

I am on the fence regarding recommending the publication of the revised manuscript.

(i) Showing that mutations co-clustered with phosphorylations are enriched in activating mutations (as described in a response to comment 5 in my previous review) is big and – for me – fulfills the requirement for publication in Nature Communications. I think this is a key result and should be highlighted in the abstract.

Authors: We thank the reviewer for suggesting the analysis and have now further highlighted this key result in main figures (see details below).

(ii) The fact that authors found 48 mutations through co-clustering that were not covered in the mutation-only clustering is interesting and should be included in the manuscript.

Authors: We have now added this analysis to the **Results**,

“We also compared the mutations in the hybrid clusters to those found in a clustering analysis using only TCGA MC3 mutations, which contained 9,403 clustered mutations. Among the 2,938 mutations found in the 474 hybrid clusters, we found only 48 mutations not found by mutation-only clustering. The list of 48 mutations contained new mutations of interest in PDE1B (5 mutations), SRSF7 (4 mutations), and PTPN12 p.S275F/C that co-localized with p.S275 and co-clustered with p.S39/p.T40 (**Supplementary Table 2**).”

(iii) I think that the analysis in combination with the DepMap data is not very clear. Kinase expression levels are not necessarily correlated with an increased kinase activity. For this analysis I rather thought of: (a) determining driver-mutations enriched in cell lines of a cancer type, (b) selecting for genes carrying driver mutations that are also show an increased fitness effect for these cell lines, (c) check if kinases associated with

co-clustering phosphosites on these driver genes also are showing a significantly enriched essentiality for cell line models of this cancer type.

Authors: We agree with the reviewer that directly assessing the cancer dependency in cell lines with co-clustered mutations vs. those with other mutations would be a more direct test. The DepMap expression-driven analysis of kinases was also removed due to the removal of the kinase regulation section (see comments for Reviewer #3). Instead, we now added this direct comparative analyses in the **Results**,

“Next, we sought to test whether the co-clustered mutations may confer genetic dependency to the mutated cancer cells. In this case, cancer cells with co-clustered mutations would show higher vulnerability in a CRISPR knockout screen targeting the mutated genes than cells with other mutations. To test this hypothesis, we utilized data using characterized by the CRISPR-knockout screens in the Cancer Dependency Map (DepMap) project²⁷, where a negative CERES dependency score indicates genetic dependency of the cancer cell. Within each of the 27 tested lineages, we carried out a Wilcoxon Rank Sum test between the cell lines with co-clustered missense mutations versus those with other missense mutations (**Methods**). Strikingly, cancer cell lines with co-clustered mutations showed significantly higher dependency (or more vulnerability upon genetic knockout) than those with missenses in 14 lineages (FDR < 0.05), most notably lung, colorectal, skin, pancreas, and gastric cancer cells (FDR \leq 3.3E-7, **Figure 5E**). We also obtained similar results when comparing cell lines with co-clustered missense mutations versus other non-synonymous mutations (**Supplementary Table 10**). Overall, these analyses showed that co-clustered mutations adjacent to phosphosites are enriched for activating events and highlight genetic vulnerability of cancer cells.”

(iv) My question on the bimodal nature of the distribution shown in Fig. 1B, and why the extend of the bimodality is different between the HotPho data and the simulated data, and how this could be used to filter for interesting co-clusters, has been mainly ignored in the rebuttal although it also was brought up by another reviewer.

Authors: We have now more closely examined the distribution of cluster closeness score (cc). Using a binned histogram with 200 bins (now added to **Figure S1**), we can visualize the distribution contained multiple modes in both simulated and observed mutation/phosphosite co-clusters. Some apparent alleys of simulated cluster scores are at $\log_{10}(Cc)$ of -0.1~0, 1.9~2.0, etc., and which minima are found is dependent on the number of bins used in the probability density function coupled with a second derivative analysis to find turning points (at 200 bins we found 56, potentially too many for defining a single threshold). The figure also showed that the current threshold of top 5% cc score (grey line) has a high specificity of 97.4% and the ROC curve analyses in **Figure S1A** showed the sensitivity/specific at the current score. If the reader wanted to be more

stringent and select a threshold at another local minima, the $\log_{10}(C_c)$ of 2 would retain the top 2.278% of the hybrid clusters. The new analysis is now combined with the previous texts in a standalone paragraph in **Results**,

“We conducted a multitude of analyses to investigate the modality in the score distribution and the implication of using the 5% threshold. First, while this threshold (cluster closeness score = 2.56) may permit false-positives if the simulated phosphosites only contain negatives, we observed many of the clusters containing activating or recurrent mutations with cluster closeness scores close to the threshold (**Supplementary Table 3**). It is possible that the spatial distribution of cancer mutations and commonality phosphosite residues (i.e., serine, threonine, and tyrosine) is not random and thus retaining these additional hybrid clusters is needed to minimize false-negatives. Second, to resolve possible reasons underlying the multi-modal distribution of cluster closeness scores, we compared the score distributions for 299 mutation-enriched cancer driver genes¹³ versus other genes. While hybrid clusters involving driver genes showed a higher density at the higher-score mode, driver gene status did not guarantee high scores (**Supplementary Fig. 1B**). The 5% score threshold showed a sensitivity = 0.17 and specificity = 96.0% in distinguishing hybrid clusters with driver genes (**Supplementary Fig. 1C**). Finally, we examined the score distribution using 200 bins on both the simulated vs. observed clusters, finding multiple peaks and alternative thresholds, for example, thresholding using one of the higher local minima retained only the top 2.28%, or the top 216 clusters (**Supplementary Fig. 1D**). Cluster closeness scores for all identified clusters are provided herein to prioritize a more stringent set of clusters (**Supplementary Table 1**).”

(v) Most of the work on addressing suggestions of the reviewers from the first round was mainly added in the form of Supp. Figures. As the (main) figures are the windows to a scientific publication, I am not sure this very useful additions and deeper analyses will benefit the readership in an optimal manner.

Authors: We have now added the plots illustrating mutations' co-clustering status show additional predictive value for activating mutations beyond mutation recurrence as the new **Figure 5C/D**, and the comparison of genetic dependencies of DepMap cell lines in the new **Figure 5E**. In expanding this figure, we moved the observations of the co-clustered phosphosites in primary tumours into a stand-a-lone **Figure 6**. We also note for the reviewers the extensive updates on Supplementary Tables and Figures (see change-tracked manuscript).

I would like the authors to address points (ii) – (v) before fully supporting the publication of the manuscript in Nature Communications.

REVIEWERS' COMMENTS

Reviewer #3 (Remarks to the Author):

The authors have removed the problematic section in the revised manuscript. This addresses my criticism of this section.

The fact that this work is an incremental advance over existing literature remains, and it's up to the editor whether this is acceptable for Nat Comm.

Reviewer #4 (Remarks to the Author):

The authors have worked on all comments that I have brought up.

Comments (i) and (ii) were fully solved.

To my slight disappointment, the authors have removed the entire chapter discussed in my comment (iii) – in part because of the criticism from another reviewer. I thought my suggestions might lead to an interesting connection from mutations through co-clustered phosphorylation sites to the essentiality of kinases independent of their mutation status.

My comment (iv) was on the bimodularity of the cluster closeness score distribution shown in Fig. 1B. The authors have looked a little bit into this, though I do not think that we are any closer to understanding the modularity.

Some of the changes the authors added in the second and third versions of the paper are now included in the main figures.

The main message of the paper is that 3D co-clustering of phosphorylation sites and mutations predicts higher dependencies on the mutated proteins, and I think it is a strong enough message to make the manuscript suitable to be published in Nature Communications.

REVIEWERS' COMMENTS

Reviewer #3 (Remarks to the Author):

The authors have removed the problematic section in the revised manuscript. This addresses my criticism of this section.

The fact that this work is an incremental advance over existing literature remains, and it's up to the editor whether this is acceptable for Nat Comm.

Authors: We thank the reviewer the constructive review.

Reviewer #4 (Remarks to the Author):

The authors have worked on all comments that I have brought up.

Comments (i) and (ii) were fully solved.

To my slight disappointment, the authors have removed the entire chapter discussed in my comment (iii) – in part because of the criticism from another reviewer. I thought my suggestions might lead to an interesting connection from mutations through co-clustered phosphorylation sites to the essentiality of kinases independent of their mutation status.

My comment (iv) was on the bimodularity of the cluster closeness score distribution shown in Fig. 1B. The authors have looked a little bit into this, though I do not think that we are any closer to understanding the modularity.

Some of the changes the authors added in the second and third versions of the paper are now included in the main figures.

The main message of the paper is that 3D co-clustering of phosphorylation sites and mutations predicts higher dependencies on the mutated proteins, and I think it is a strong enough message to make the manuscript suitable to be published in Nature Communications.

Authors: We thank the reviewer the constructive review.